# Learning shared neural manifolds from multi-subject FMRI data

## Abstract

Functional magnetic resonance imaging (fMRI) is a notoriously noisy measurement of brain activity because of the large variations between individuals, signals marred by environmental differences during collection, and spatiotemporal averaging required by the measurement resolution. In addition, the data is extremely high dimensional, with the space of the activity typically having much lower intrinsic dimension. In order to understand the connection between stimuli of interest and brain activity, and analyze differences and commonalities between subjects, it becomes important to learn a meaningful embedding of the data that denoises, and reveals its intrinsic structure. Specifically, we assume that while noise varies significantly between individuals, true responses to stimuli will share common, low-dimensional features between subjects which are jointly discoverable. Similar approaches have been exploited previously but they have mainly used linear methods such as PCA and shared response modeling (SRM). In contrast, we propose a neural network called MRMD-AE (manifold-regularized multiple decoder, autoencoder), that learns a common embedding from multiple subjects in an experiment while retaining the ability to decode to individual raw fMRI signals. We show that our learned common space represents an extensible manifold (where new points not seen during training can be mapped), improves the classification accuracy of stimulus features of unseen timepoints, as well as improves cross-subject translation of fMRI signals. We believe this framework can be used for many downstream applications such as guided brain-computer interface (BCI) training in the future.

## 1 Introduction

Brain activation dynamics, as measured by fMRI, exist in an extremely high-dimensional collection space and contain high levels of noise. Noise sources include measurement imprecisions and physiological factors such as a subject's blood pressure variability, head motion, and respiration (Brooks et al., 2013; Mumford & Poldrack, 2007). Moreover, even after accounting for the idiosyncrasies in noise, neural responses evoked by the same stimulus can vary greatly between different subjects. This makes using fMRI to extract meaningful signals or general trends that are task-relevant quite difficult. Experiments involving fMRI are expensive and usually involve pilot studies conducted in a smaller scale with simpler stimuli and on a smaller group of subjects. Thus, it is desirable to have a method that can learn common trends from pilot data and can generalize to a larger group of subjects or more complex tasks.

Like for other high dimensional biomedical data, it is broadly acknowledged that neural activity actually lies in a lower-dimensional latent space. Moreover we hypothesize that several features of these latent spaces are shared across subjects given that measurements are usually task-based and are taken when subjects experience the same stimuli. Dimensionality reduction methods have been used to discover such low-dimensional spaces and facilitate understanding the underlying brain activities. Methods such as principal component analysis (PCA) and factor analysis have been the common choice (Smith et al., 2014). However, these linear methods are sensitive to noise and restrict interactions in the latent space. Nonlinear dimensionality reduction algorithms are shown to better capture the geometry of high-dimensional biological data (Moon et al., 2019; Van der Maaten & Hinton, 2008) and account for dynamics in neural activities (Gao et al., 2020). PHATE (Moon et al.,

2019), a diffusion based manifold learning method, was shown to capture both global and local geometry of complex biological data, which appears to be a good candidate to model fMRI data.

Using a nonlinear manifold such as PHATE to reduce the dimensionality of data is appealing as it can extract meaningful signals, denoise the data, and accelerate downstream analysis. However, unlike a method like PCA which learns a projection operator that can be applied to new data, manifold learning methods are fixed to the input data and do not naturally extend to new data from new tasks or from new subjects. To extend an existing manifold to out-of-sample data, landmark interpolation or Nyström extensions are commonly used. However it is usually unsatisfactory as shown by decremented performance(Appendix Fig. 1). To tackle these shortcomings, we turn to neural networks such as autoencoders that provide nonlinear dimensionality reduction via a learned parametric function that is readily applicable to new data. While our key goal is learning an informative low-dimensional data manifold shared across multiple subjects, this is not something automatically done by autoencoders. First, there is nothing enforcing autoencoder latent embeddings to respect data manifold geometry, and indeed they often just spread out points for ease of decoding. Second, individual variation often dominates embeddings which prevents a common space to be automatically learned.

To address these issues, we propose a manifold-regularized *multi-decoder* autoencoder (MRMD-AE) that can process fMRI data from a group of subjects and extract common latent space representations that respect individual data geometry. Key features of the MRMD-AE include: (a) A common encoder that projects fMRI data from any subject on to a shared latent space, and subject-specific decoders that learn to reconstruct data for each subject faithfully. (b) A manifold-geometry regularization of the latent space based on a precomputed PHATE embedding. (c) Penalties for distances between individual patient encodings of common stimuli to ensure that the latent space is not split into individual embeddings.

We show results on three types of tasks in two different datasets. First, we show that we are able to extend the manifold embedding to new timepoints of data not used in training. This shows the *extensibility* of our manifold in contrast to non-neural network based manifold learning methods. Second, we show that our latent space improves the ability to classify or infer stimulus features based on subject fMRI measurements. Surprisingly, the multiple decoder improves classification accuracy over a common decoder, showing that this framework allows the network to separate common from individual variations. Finally, we show that even untrained, translation between subjects has increased accuracy on withheld timepoints. Further, we show highly improved cross subject translation after training for this task.

## 2    RELATED WORK

**Manifold Learning**    Manifold learning assumes that high-dimensional ambient data $X \in \mathbb{R}^d$ lies on a low dimensional manifold. Given a set of datapoints measured in the ambient space, assumed to be sampled from a manifold $\mathcal{M}$, manifold learning optimizes a low dimensional Euclidean space that encodes the intrinsic geometry of $\mathcal{M}$ and the original data. Popular manifold learning methods include diffusion maps (Coifman & Lafon, 2006), Laplacian Eigenmaps (Belkin & Niyogi, 2003) and PHATE (Moon et al., 2019). We note that multiscale manifold learning methods (Wolf et al., 2012; Brugnone et al., 2019; Kuchroo et al., 2020) can be used to consider multiple manifolds that capture relations and structure in data at different resolutions. Other methods, such as tSNE (Van der Maaten & Hinton, 2008) and UMAP (McInnes et al., 2018), relax the manifold assumption to focus on neighborhood preservation in low dimensions. While these are often considered as variations of manifold learning, they are mostly suitable for visualization of clustered data that does not necessarily have global structure, and typically do not preserve overarching trends or relations between data regions (see, e.g., Moon et al., 2019; Gigante et al., 2019).

**SAUCIE**    In (Amodio et al., 2019), SAUCIE (Sparse Autoencoder for Clustering, Imputation, and Embedding) is an autoencoder-based generative model that manipulated internal representations to force the network to produce specific desired transformations in the data. By imposing these structure on the latent representations of data, it can quickly extract information about datasets more massive than alternative approaches can easily handle. Further, this framework establishes that multiple regularizations, in conjunction, can be used to encourage interpretable latent representations that accentuate biologically reliable relations in data coming from multi-patient cohorts, as demonstrated, e.g., in immune response profiling (Zhao et al., 2020). Inspired by the success of this approach in

immunology, we apply a similar multi-objective neural network approach here to process multi-subject fMRI data and alleviate some of the main challenges associated with this data type.

**SRM**    Shared response modeling (SRM) is a linear method to account for differences in functional topographies in fMRI data in a low-dimensional space. SRM uses a singular value decomposition approach to learn a set of low-dimensional signals (shared response) common to multiple subject's fMRI data during the same experiment. SRM learns subject-specific transformation to align subject data with the shared model using an orthonormally-constrained weight matrix, similar to that of PCA (Chen et al., 2015).

## 3  PRELIMINARIES

**Manifold learning and dimensionality reduction with PHATE**    To capture the complex interaction in the latent space and balanced global and local structures of highly noisy fMRI data, we use Potential of Heat-diffusion for Affinity-based Transition Embedding (PHATE) (Moon et al., 2019), a manifold learning method based on diffusion geometry (Coifman & Lafon, 2006). PHATE computes the diffusion operator which is the probability transition matrix $P$ between data point pairs via an $\alpha-$decay kernel. Then by raising $P$ to the power of $t$, the PHATE optimal diffusion time scale, it simulates a $t-$step random walk over the data graph. Diffusion potential distances $D$ is extracted from $P^t$, which accounts for both global and local data geometry. Finally, metric multi-dimensional scaling (metric MDS) (Abdi, 2007), is applied to $D$ to get the PHATE embedding of input data. PHATE has been shown to effectively capture geometry of high-dimensional biological data and was shown to preserve manifold structure in fMRI data in (Rieck et al., 2020). We utilize the extensible framework of PHATE and use PHATE embedding of the training data as regularization applied to the manifold embedding layer as shown in Fig. 1. Note that we use PHATE over other popular dimensionality reduction methods such as tSNE or UMAP because of its established ability to retain manifold and trajectory structure in biomedical data (Moon et al., 2019; Kuchroo et al., 2020; Chung et al., 2020; Pappalardo et al., 2020).

**Geometry-regularized Autoencoders (GRAE)**    The latent space of the vanilla autoencoders are usually difficult to interpret and not necessarily reflect data geometry (see, e.g., Mishne et al., 2019; Moor et al., 2020; Duque et al., 2020). In order to encourage meaningful data geometry to emerge, or be retained, in latent representations, the training process can be enhanced with appropriate regularizations, often derived from manifold learning principles. For example, Jia et al. (2015) and Yu et al. (2013) propose regularization terms that penalize inaccurate preservation of neighborhood relationships in the ambient space. Recent work by Duque et al. (2020) has shown that the geometry extracted by PHATE can be leveraged to formulate Geometry-Regularized Autoencoders (GRAE) that faithfully capture the instrinsic data geometry, while demonstrating advantages in extendibility and invertibility compared to previous approaches. In this work we adapt this approach to fMRI data in conjunction with adjusted architecture and additional regularizations that address further challenges encountered in these settings.

**Cross-subject batch effects in fMRI**    Relative to other neuroimaging methods, fMRI is noninvasive, minimal risk, and has broad coverage with high spatial resolution. fMRI experiments commonly use naturalistic stimuli such as movies to probe an array of cognitive mechanisms in conditions that closely mimic the human experience of the world. However, the data generated by such experiments are notoriously high-dimensional and complex, as fMRI is rife with noise from sources like the scanner, subject movement, and the measured blood-oxygen-level-dependent (BOLD) signal, which is a secondary effect of neural activity. This makes it difficult to aggregate data across sessions, either within or between subjects in a study. Beyond the noise sources of fMRI data, another problem with aggregating data across subjects to uncover something common about brain functioning pertains to the idiosyncratic nature of functional topographies in the brain. In many brain regions, shared stimuli will reliably evoke similar patterns of neural activity across subjects (Nastase et al., 2019). However, that signal of interest could be located in slightly different anatomical locations across subjects or otherwise obscured by noise, causing a mismatch between functional and anatomical locations for a signal. A large body of neuroimaging research centers around rectifying this problem with a variety of linear functional alignment methods by learning either high-dimensional (Haxby et al., 2011; Busch et al., 2021) or reduced-dimensional (Chen et al., 2015) common spaces. A solution to this issue must consider both the complexity of the data at the individual subject level, where there is biological and environmental noise, and the broader cross-subject representational idiosyncrasies.

## 4    MULTI-DECODER MANIFOLD-REGULARIZED AUTOENCODER

### 4.1    PROBLEM SETUP

Let $X = \{X_k : k = 1, \ldots, m\}$ be a set of common stimuli for $m$ timepoints, and let $Y_j = \{Y_{j,k} \in \mathbb{R}^q : k = 1, \ldots, m\}$, $j = 1, \ldots, n$ [1], be observed fMRI neuronal activations for $n$ subjects over these stimuli. We seek to learn a low dimensional latent space $\mathcal{M} \subset \mathbb{R}^d$ (where $d \ll q$) with associated map $f : \mathbb{R}^q \to \mathbb{R}^d$ and inverse maps $g_j : \mathbb{R}^d \to \mathbb{R}^q$, $j = 1, \ldots, n$, such that: (a) The mappings corresponding to each stimulus $X_k$ in subjects $Y_i$ and $Y_j$ are close to one another, i.e., $\|f(Y_{i,k}) - f(Y_{j,k})\|_2 \le \epsilon$. (b) Stimulus features (for each $X_k$) can be computed by a function $f(Y_{j,k}) \mapsto f'(X_k)$ of the low dimensional latent space. (c) Stimulus features can be inferred in an unseen/untrained subject $Y_{n+1}$ from the same latent space via $f(Y_{n+1,k}) \mapsto f'(X_k)$, at any timepoint $k$. (d) Subject-specific responses can be predicted on unseen stimulus $X_{m+1}$ (e.g., at held-out timepoint $m + 1$) based on the shared latent space, i.e., $g_j(f(Y_{i,m+1})) = Y_{j,m+1}$ for subjects $i, j$. Note here that each $Y_{i,k}$ is a high dimensional measurement consisting of raw activation values at hundreds to thousands of voxels depending on the respective region of interest (ROI). By contrast, we expect $f(Y_i)$ to be vastly lower dimensional. We analyzed the spectral entropy of PHATE diffusion potential of the fMRI activation data used in the following experiments and found the intrinsic dimensions of the data ranging 10 to 40 (Appendix Table 1).

### 4.2    MRDM-AE ARCHITECTURE

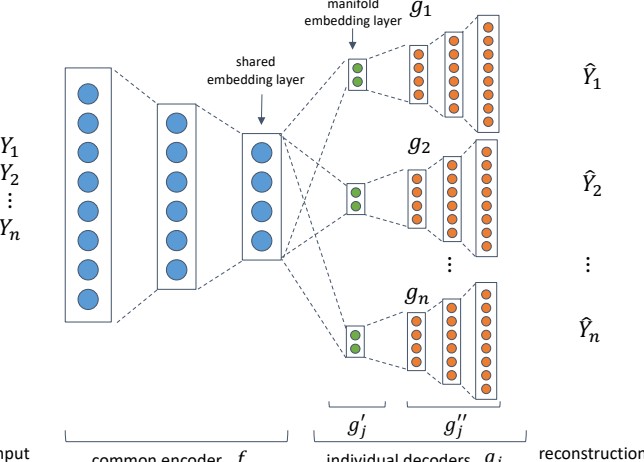

Figure 1: Multi-decoder Manifold Regularized Autoencoder

**Single Encoder – Multiple Decoders**    To achieve the properties outlined in the previous section, we propose a manifold-regularized common-encoder, multiple-decoder model (Figure 1). The common encoder $f(Y_{j,k}) = \text{Enc}(Y_{j,k})$ takes any set of neuronal activities from any subject and maps them in a many-to-one fashion to the same latent space $L$, with the same stimulus prediction. Our model features several decoders $g_j(f(Y_{\cdot,k})) = \text{Dec}[j](f(Y_{\cdot,k})) \approx Y_{j,k}$, to decode back to subject specific responses. The rationale for using a common encoder is for the model to learn a common fMRI language from which subject specific responses can be derived. While the fMRI data exhibits strong batch effect between subjects, the common encoder builds invariance in the latent space to individual noise in the encoded space. The individual decoders enable incorporating subject-dependent interpretation of the latent information from the common encoder. Thus this framework pushes individual noise into the decoding space. In particular, the decoders can learn to reconstruct the subject-specific signals and recreate batch effect to reconstruct data. The flexibility provided by individual decoders allow the single common encoder to only retain input-dependent variability between subject responses that is sufficient for the decoders to reconstruct the measurements. This base model is trained with a reconstruction error penalty for each subject seen at training time.

$$L_{f,g_1,\ldots,g_n}^{\text{reconstruction}}(Y_1, \ldots, Y_n) = \sum_{j,k} \|Y_{j,k} - g_j(f(Y_{j,k}))\|^2$$

---

[1] During preprocessing, functional images were aligned to the common template based on anatomical landmarks and one time point of a fMRI region of interest is a vector of length equal to the number of voxels.

**Embedding Alignment Penalty**   Despite the common encoder (and individual decoders), the network may learn to separate subject-specific responses into their own latent sub-spaces for ease of decoding. To counteract this tendency we explicitly push the latent encodings of individual responses to common stimuli together by penalizing Euclidean distances in the encoded space between timepoints corresponding to common stimuli. Therefore, we implement the embedding alignment penalty as $L_{f,g_1,\ldots,g_n}^{\text{alignment}}(Y_1,\ldots,Y_n) = \sum_{i\neq j,k} \|f(Y_{i,k}) - f(Y_{j,k})\|^2$.

**Manifold Regularization Penalty**   In order to ensure a smooth embedding space that enables faithful interpolation, inference, and extendibility to new data, a geometric regularization should be applied to ensure the latent space learned by the autoencoder follows the data geometry. To this end, within each decoder, we apply a regularization in the bottleneck layer using the manifold embedding $\mathcal{P}_j$ that are learned with PHATE (Moon et al. (2019)) using the train data for each subject $j$. We note that the strong subject batch effects in fMRI data make the task of directly learning a subject-independent manifold with PHATE challenging. Indeed, without careful consideration, it often yields a group fMRI data geometry with variance dominated by batch effects, which would then submerge stimulus-related variance that is of actual interest. We therefore bypass this issue here by only considering individual-subject manifolds and letting the combined autoencoder deduce a latent representation that can easily be mapped to them. Thus, as shown in Figure 1, we split each decoder into $g_j = g_j' \circ g_j''$, where intuitively we expect the first part $g_j'$ to "implement batch effects" going from the common latent representation to a subject-dependent one that captures well the intrinsic structure of their stimuli responses. It is to this first layer of $g_j'$ we apply the manifold regularization. The second part $g_j''$ will then map this intrinsic (manifold) representation to the observable measurements to finish reconstruction. In this way, we obtain an intermediary representation after the first layer of each decoder (where the single-layer limitation restricts the implemented "batch effects" to be relatively simple) that should match the PHATE embeddings, implemented in training via the manifold regularization as $L_{f,g_1',\ldots,g_n'}^{\text{geometric}}(Y_1,\ldots,Y_n) = \|g_j'(f(Y_{j,k})) - \mathcal{P}_j(Y_{j,k})\|^2$.

**Combined Loss**   To combine three loss terms together, we introduce coefficients $\lambda, \mu$ controlling the amount of alignment and geometric regularization (correspondingly) applied to the hidden layers of the MRMD-AE, yielding the total loss

$$L = \sum_{j,k} \|Y_{j,k} - g_j(f(Y_{j,k}))\|^2 + \lambda \sum_{i\neq j,k} \|f(Y_{i,k}) - f(Y_{j,k})\|^2 + \mu \|g_j'(f(Y_{j,k})) - \mathcal{P}_j(Y_{j,k})\|^2 \quad (1)$$

optimized over network functions $f, g_j = g_j' \circ g_j'', j = 1,\ldots,m$, parameterized via neural networks and organized together as illustrated in Fig. 1.

**Additional Cross-Subject Training Penalty**   The multiple decoder framework allows us to add carefully designed loss penalties for specific tasks. For example, we can explicitly train for cross-subject translation. We achieve the purpose by adding the following regularization to encourage translation,

$$L_{f,g_1,\ldots,g_n}^{\text{translation}}(Y_1,\ldots,Y_n) = \sum_{j,k} \sum_{l\neq j} \|Y_{j,k} - g_j(f(Y_{l,k}))\|^2. \quad (2)$$

In most of the following experiments this cross-subject penalty was not used, as we did not aim to train for translation. In the last section of Results we explore the use of this penalty to facilitate learning to translate between subjects' fMRI activities.

## 5   RESULTS

To empirically validate MRDM-AE, we test it on tasks of manifold extension, stimulus classification, and cross-subject translation. The manifold-extension tasks test the ability of MRDM-AE to place points not seen during training appropriately within the learned data manifold in the embedded space. The classification tasks analyze the ability of MRDM-AE to capture stimulus-relevant signal from fMRI data, i.e., features of the movies from voxel activations. Cross-subject classification and translation tasks test the ability of MRDM-AE to capture a shared latent space that can be used for common encoding and individual decoding. These tasks are tested on two different datasets comprising of subjects watching different movies. We compare MRDM-AE to several baselines including dimensionality reduction methods (PCA, PHATE), the shared response model (SRM) commonly used in fMRI data, as well as ablations of our autoencoder method.

**Datasets**    The Sherlock dataset is an open-access dataset of 16 participants watching a 48-minute clip of the BBC television series "Sherlock." Data was collected in two runs of 946 and 1030 timepoints (repetition time; TRs) respectively. This data was downloaded from the DataSpace public repository. For full details on the stimulus, see the original publication of this dataset (Chen et al. (2017)). Annotations for the *Sherlock* movie stimulus were released with Vodrahalli et al. (2018) and available freely on Github. These annotations were created by humans who viewed the film and labeled each timepoint on a variety of metrics. In our classification tasks, we used two sets of binary labels that correspond with visual and auditory aspects of the stimulus: whether a given timepoint featured an indoor or outdoor scene (Indoor/Outdoor) and whether or not the timepoint featured music (Music Present).

As a second testbed, we used the StudyForrest open-access dataset downloaded from DataLad; full details for this can be found at the original publication of the movie dataset (Hanke et al. (2014)) and the localizer extension dataset (Sengupta et al. (2016)). Here we included data from 15 participants who completed both the movie-viewing and functional localizer tasks, where participants viewed 24 unique grayscale images of objects from 6 categories (faces, bodies, houses, small objects, outdoor scenes, and scrambled images) in a separate session to localize brain regions that reliably respond to a specific category. In the movie task, participants watched a 2-hour version of the movie *Forrest Gump*. In a series of classification tasks, we use labels for each timepoint of the movie indicating the time of day (2 levels), and the flow of time in the narrative (4 levels), annotated in (Husler & Hanke (2016)), and image categories for the localizer data (6 levels).

**Autoencoder Hyperparameters**    All autoencoder based models have three fully connected layers in the encoder and the decoder(s), with a bottleneck latent space layer in between. The hidden neurons have the configurations of 256-128-64-20-64-128-256. Leaky ReLU activations are applied on all of the layers. We used the Adam optimization algorithm with a learning rate of 0.001, and a batch size of 64. The neural networks were trained for 4000 epochs. When regularization is used, $\lambda$ and $\mu$ are either 0.1 or 0.01 and the same set of values are used across subjects for a specific task.

**Model Comparison**    We compare our proposed model with a standard autoencoder (AE) with the same hidden dimensions as the proposed MRMD-AE, but with a single decoder for all subjects. We refer a standard autoencoder with manifold regularization applied to the bottleneck layer as MR-AE.

**Dimensionality Reduction Methods**    Non-linear dimensionality reduction methods such as PHATE (and other methods such as t-SNE and UMAP) typically provides fixed latent representations for the input data, and do not provide a natural embedding function that can project new data to an existing manifold. PHATE preserves manifold structure while tSNE and UMAP usually shatter data trajectories to create clusters even when they do not exist ( Moon et al. (2019), Kuchroo et al. (2020), Rieck et al. (2020)). We therefore compare with an adaption of PHATE to interpolate new datapoints via landmark approach used by Moon et al. (2019), a subsampling method that was used and shown to be effective in improving scalability of PHATE. We also compare with linear PCA approach. For within-subject analyses, the dimensionality reduction was learned for a subject on one half of the dataset and applied to unseen timepoints. For cross-subject analyses, the dimensionality reduction was learned on all but one subject and applied to the held-out subject, and repeated holding out each subject once. For cross-subject analyses, we also compared to SRM by learning a reduced-dimensional shared signal on data from all but one subject and evaluating the fit of the shared space on the held-out subject, repeating for all subjects.

**Place Future fMRI Time Points on Learned Manifold**    We first show that MRMD-AE achieves efficient and accurate manifold extension by placing new fMRI time points at the right positions on learned manifold. Consider the fMRI time series contain two parts, including the first $m_{train}$ TRs that we receive as pilot data with which we learned individual manifolds $\mathcal{P}_{train}$, a $m_{train} \times d$ matrix, with each row being the PHATE coordinates of one TR. We would like to faithfully extend these learned manifolds to subsequently receive test data of future $m_{test}$ fMRI TRs, so that the extended manifold $\mathcal{P}_{test}$ closely resemble the corresponding time points ground truth manifold $\mathcal{P}$, that is learned from all $m_{train} + m_{test}$ TRs.

Since the manifold coordinates are fixed to the input data, we cannot directly compare the $\mathcal{P}_{test}$ with $\mathcal{P}[TR_{test}]$, the corresponding rows of $\mathcal{P}$. Instead, we first compute the manifold distance matrices between each test and train time point pairs, where $\hat{\mathcal{D}}(i,j) = ||\mathcal{P}_{test}(i,) - \mathcal{P}_{train}(j,)||$, and the ground truth $\mathcal{D}(i,j) = ||\mathcal{P}(i,) - \mathcal{P}(j,)||, i \in TR_{test}, j \in TR_{train}$. Then we evaluate the similarity using the MSE between $\hat{\mathcal{D}}$ and $\mathcal{D}$. Fig.2 shows that MRMD-AE achieves lower or matching MSE than the landmark approach, especially with lower percentages of train data. The extended manifold

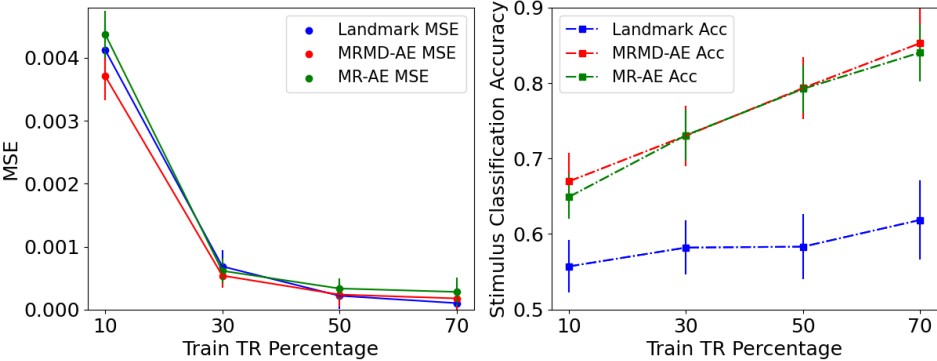

Figure 2: Comparison of extended manifold by MRMD-AE, MR-AE and PHATE Landmark on the Sherlock early-visual ROI. The MSE between extended manifold coordinates and ground truth were computed (lower is better). MRMD-AE extended manifold embeddings achieve highest classification accuracy in predicting movie stimulus labels. Error bars are standard deviation across subjects.

generated by MRMD-AE more closely resemble the ground truth. MRMD-AE also outperforms MR-AE, suggesting the advantage of per-subject decoders in reconstructing subject-specific signals.

We also use the extended manifold embedding to predict movie stimulus labels. An support vector classifier (SVC) is trained on each subject's extended manifold coordinates and the average accuracy and standard deviation via a 5-fold cross validation are reported. We found that MRMD-AE extended manifold achieves higher classification accuracies than the landmark approach, suggesting the embeddings retain more meaningful signals that can be used to recover signals of the movie stimulus (right panel of Fig. 2 and Table 1).

**Effect of Latent Space Alignment Penalty** We visualize the raw data and the embedding $f(Y_j), j = 1, 2, \ldots, 15$ of the Forrest image localizer data using PHATE, and color the plots by subject ID in Figure 3. The raw data visualization exhibits clear batch effect. By increasing $\lambda$ in Eq. 1 and penalizing latent space misalignment, the batch effect in the latent space is eliminated. We computed the average Earth Mover's Distance (EMD) between subject embedding pairs and showed improved alignment between subjects in the latent space comparing to the raw data and the improved alignment by increasing alignment penalty.

**Stimuli Classification** Our goal is to test how well our MRMD-AE generalizes to unseen times series of fMRI data. We evaluate the model by how well we can recover meaningful labels in the latent space. After training on the train half of data from all subjects, we applied the trained model on the test half of data and collected the latent space representation at the bottleneck layer. A SVC is trained within each subject on the test embedding and the average classification accuracy and standard deviation via a 5-fold cross validation are recorded. The average accuracy and standard deviation across subjects are reported.

Results are reported in table 1 for Sherlock movie annotation classification and table 2 for Forrest movie and image localizer classification. MRMD-AE effectively recovers the latent factors corresponding to the stimuli and achieve the best or second best performance in all combinations of ROI and stimulus annotations. We showed previously that a learned PHATE manifold via landmark does not generalize to out-of-sample data. Here we confirm that the classification accuracy of the test set decreases significantly from the train set. The linear projection operator learned by PCA also fails to generalize well to new data (Appendix Fig. 1 shows in-sample vs out-of-sample comparison for PCA and PHATE Landmark). MRMD-AE also outperforms per-subject individual MR-AE, suggesting the synergy between subjects offered by the shared encoder and multi-decoder framework benefits generalization and recovering meaningful signal from data. Appendix Figure 2 shows that across subjects, MDMR-AE consistently outperforms the other methods.

**Untrained Cross-Subject Translation** Our MRMD-AE framework allows integration of data from multiple subjects, thereby enabling translation between subjects. Similar as before, we trained our MRMD-AE with the train half of all subjects' data. Then we translate the test half timepoints of subject $i$ to each of the other subjects by $\hat{Y}_{i \to j} = g_j(f(Y_i))$. We evaluate the quality of translation by comparing the MSE and correlation between $\hat{Y}_{i \to j}$ and $Y_j$( 3). It should be noted that the model was not trained for translation, thus in training, the decoders did not learn to translate other subjects data. Yet, the common encoder maps the neural activities to the same latent space and builds invariance to individual noise in the encoded space, allowing the decoders to perform the translation task. SRM was

Table 1: **Sherlock within-subject manifold classification.** Within each subject's data, a latent space was learned on one half of the data and extended to the held-out timepoints. A support vector classifier was trained/tested on the latent representation of the unseen timepoints with 5-fold cross validation to predict features of the movie stimulus. The mean and standard deviation of test accuracy across subjects and folds were reported. MRMD-AE outperformed all comparison methods suggesting efficient manifold extension to new time points and better preservation of factors from the stimulus.

| ROI | Method | IndoorOutdoor | | MusicPresent | |
|---|---|---|---|---|---|
| | | Accuracy | Std. Dev. | Accuracy | Std. Dev. |
| early-visual | PCA | 0.5923 | 0.0629 | 0.5215 | 0.0247 |
| | PHATE-Landmark | 0.5681 | 0.0333 | 0.5530 | 0.0302 |
| | AE | 0.7379 | 0.0499 | 0.6840 | 0.0240 |
| | Individual MR-AE | 0.7289 | 0.0196 | 0.6746 | 0.0259 |
| | MRMD-AE | **0.7760** | 0.0520 | **0.7170** | 0.0383 |
| early-auditory | PCA | 0.5948 | 0.0606 | 0.5544 | 0.0314 |
| | PHATE-Landmark | 0.5831 | 0.0290 | 0.5665 | 0.0296 |
| | AE | 0.7285 | 0.0445 | 0.6950 | 0.0268 |
| | Individual MR-AE | 0.7213 | 0.0315 | 0.6875 | 0.0309 |
| | MRMD-AE | **0.7523** | 0.0544 | **0.7128** | 0.0381 |
| pmc | PCA | 0.5622 | 0.0640 | 0.5216 | 0.0244 |
| | PHATE-Landmark | 0.5395 | 0.0368 | 0.5342 | 0.0316 |
| | AE | 0.7080 | 0.0446 | 0.6585 | 0.0243 |
| | Individual MR-AE | 0.7096 | 0.0207 | 0.6600 | 0.0265 |
| | MRMD-AE | **0.7318** | 0.0589 | **0.6845** | 0.0403 |

Table 2: **Forrest within-subject manifold classification.** For Flow of Time and Time of Day classifications, latent spaces were learned on one half of the movie timepoints and then extended to the held-out timepoints. For the Localizer classification, latent spaces were trained on the entirety of the movie watching dataset and then extended to the localizer dataset, which was collected in a separate session. Results are reported as the average and standard deviation across subjects and cross validation folds.

| ROI | Method | Flow of Time | | Time of Day | | Localizer | |
|---|---|---|---|---|---|---|---|
| | | Acc. | Std. Dev. | Acc. | Std. Dev. | Acc. | Std. Dev. |
| early-visual | PCA | 0.4590 | 0.0186 | 0.6406 | 0.0322 | 0.3965 | 0.0464 |
| | PHATE-Landmark | 0.2434 | 0.0135 | 0.5266 | 0.0233 | 0.3863 | 0.0182 |
| | AE | 0.5547 | 0.0210 | 0.6025 | 0.0298 | **0.4542** | 0.0561 |
| | Indiv. MR-AE | 0.5771 | 0.0242 | 0.6695 | 0.0197 | 0.4321 | 0.0484 |
| | MRMD-AE | **0.5845** | 0.0192 | **0.6912** | 0.0216 | 0.4468 | 0.0683 |
| late-visual | PCA | 0.4690 | 0.0167 | 0.6450 | 0.0287 | **0.4442** | 0.0187 |
| | PHATE-Landmark | 0.2540 | 0.0219 | 0.5244 | 0.0236 | 0.3817 | 0.0104 |
| | AE | 0.5613 | 0.0220 | 0.6278 | 0.0270 | 0.4153 | 0.0491 |
| | Indiv. MR-AE | 0.5773 | 0.0227 | 0.6659 | 0.0229 | 0.4023 | 0.0392 |
| | MRMD-AE | **0.5837** | 0.0209 | **0.6816** | 0.0237 | 0.4255 | 0.0514 |

Table 3: Cross-subject Translation Comparison when MRMD-AE is not trained for translation. The translation accross subjects $\hat{Y}_{i \to j} = g_j(f(Y_i)), i \neq j$ is compared with the target subject $Y_j$. Lower MSE and higher correlation indicate better translation. MRMD-AE trained with latent space alignment penalties ($\lambda > 0$ in Eq. 1) achieves better translation.

| ROI | Method | MSE | | Correlation | |
|---|---|---|---|---|---|
| | | Reconstruction | Translation (untrained) | Reconstruction | Translation (untrained) |
| early-visual | PCA | 0.1548 | 1.3903 | 0.9193 | 0.2275 |
| | SRM | 0.1640 | **1.3504** | 0.9142 | 0.2327 |
| | MRMD-AE ($\lambda = 0$) | 0.1797 | 1.4321 | 0.9058 | 0.2235 |
| | MRMD-AE ($\lambda > 0$) | 0.1933 | 1.3626 | 0.8981 | **0.2517** |
| pmc | PCA | 0.0752 | 1.5354 | 0.9616 | 0.1928 |
| | SRM | 0.0766 | 1.5130 | 0.9609 | 0.1951 |
| | MRMD-AE ($\lambda = 0$) | 0.0848 | 1.5540 | 0.9567 | 0.1900 |
| | MRMD-AE ($\lambda > 0$) | 0.1647 | **1.4795** | 0.9139 | **0.1977** |

developed to achieve this goal by mapping all subjects to a shared response, then project to individual spaces. We show that our model achieves best translation performance on various brain ROIs in at least one metric. By incorporating latent space alignment penalty ($\lambda > 0$ in Eq. 1), MRMD-AE learns a more aligned shared latent space and allows for better translation across subjects. More experimental details can be found in the supplementary materials.

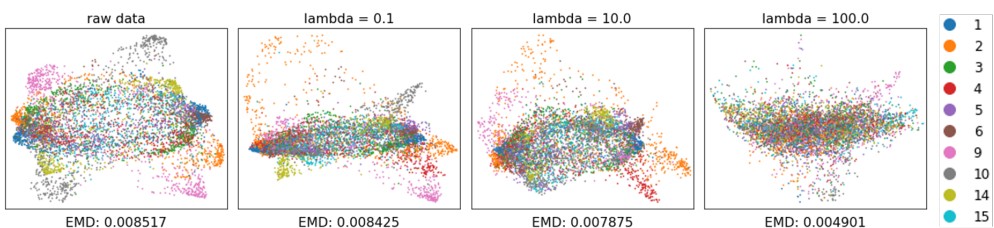

Figure 3: PHATE visualization of raw data and latent representation $f(Y_j), 1 \leq j \leq 15$, colored by subject. The raw data exhibits clear batch effect while the latent space is better aligned between subjects. The incorporation of higher levels of latent space alignment penalty by increasing $\lambda$ in Eq. 1 better removes batch effect in the shared space, demonstrated by decreasing EMD.

**Trained Cross Subject Translation**   Our MRMD-AE framework provides flexibility to specialize in different tasks by incorporating carefully designed loss penalties. By adding a cross-subject training penalty in Eq. 2 to the combined loss, we explicitly train our model for translating between subjects. We show in Figure 4 the improved translation performance by using different combinations of loss penalties. Combining the alignment penalty with translation penalty achieves the lowest translation MSE.

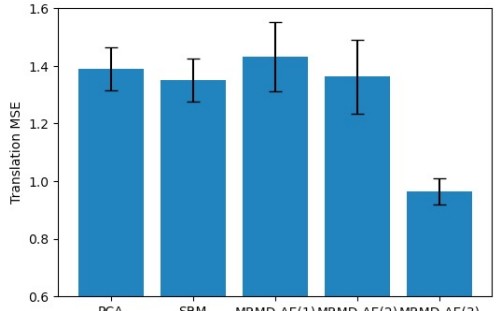

| | Alignment Penalty | Translation Penalty |
|---|---|---|
| MRMD-AE(1) | ✗ | ✗ |
| MRMD-AE(2) | ✓ | ✗ |
| MRMD-AE(3) | ✓ | ✓ |

Figure 4: Cross subject translation MSE comparison. Combinations of latent space alignment penalty and translation penalty were used. MRMD-AE with better aligned latent space achieve better translation MSE. We also trained the model explicitly for translation by incorporating the translation loss, achieving the best translation performance with the lowest translation MSE.

## 6 DISCUSSION

Extracting meaningful brain mechanisms shared across subjects is a daunting task, chiefly due to stimulus-independent noise and the challenge of comparing subjects' activity pattern in a common reference space.To address this, we proposed MRDM-AE, a multi-decoder manifold-regularized autoencoder that takes fMRI data from a group of subjects as input, first projects it to a common latent space, then to subject-specific decoders. We showed that the subject-specific decoders enabled the autoencoder to decouple shared information from individual variations such that the shared latent space offered higher-accuracy classifications of underlying stimuli, and better manifold extensions to unseen data. Finally, we showed that the individual decoders can potentially be used for subject-to-subject translation, setting up fMRI response predictions of subjects to unseen data based on training and calibration on common stimuli. Overall, we show the manifold learning and structural priors bring in improve the state of the art in shared latent space learning in fMRI from many subjects.

## REPRODUCIBILITY STATEMENT

The results presented in the paper are reproducible. We will provide a link to an anonymous repository to the reviewers and area chairs. The content of the link will contain complete instructions on how to run the code to reproduce the results presented for the paper. In addition, all details for the models and training procedures are described in the main paper under Sec. 4 and the Appendix.

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

# A Supplementary Materials

**fMRI pre-processing and region identification**    After all standard fMRI preprocessing steps, functional images from both the Sherlock and Forrest datasets were aligned to the common MNI template brain based on anatomical landmarks. In this anatomical template, we defined several regions of interest (ROIs) to target in our analyses: the early visual cortex, late visual cortex, early auditory cortex, and the posterior medial cortex (PMC). We selected these regions as they robustly respond to audiovisual stimuli (early visual, late visual, and early auditory) and PMC was targeted in the original *sherlock* analyses for its memory involvement (Chen et al. (2017)). We extracted timeseries data from the voxels in each of these regions (early visual: 307 voxels, late visual: 1008 voxels, early auditory: 1018 voxels, PMC: 481 voxels) and collapsed the spatial structure within this ROI into a [timepoints, voxels] matrix of activity for each voxel across time. Within this matrix, we z-scored each voxel across its timeseries to account for differences across voxels in mean activation.

**Dimension reduction methods on out-of-sample data**    Dimensionality reduction methods do not generalize to out-of-sample data with decremented performance when extending trained models to new data points. In Figure 1 we show that the stimuli classification accuracy decreases when PCA and PHATE (via landmark approach) are applied to out-of-sample data. Experiment was conducted on Sherlock early-visual ROI data.

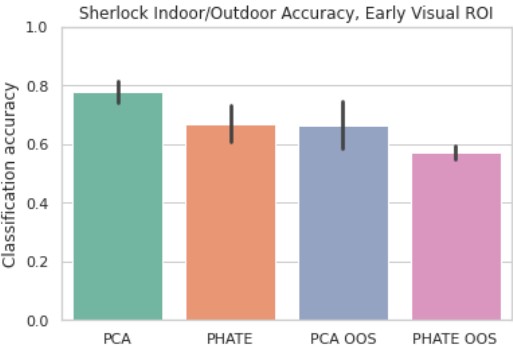

Appendix Figure 1: Classification accuracy comparison. PCA and PHATE both fail to generalize to out-of-sample data, shown as decremented accuracy. Landmark approach was used to extend learned PHATE embedding to new data.

**Stimuli classification experimental details**    For 5-fold cross validation , we proceed following the standard procedures. We split the dataset into 5 groups at random, and for each group we take as a hold out or test dataset and use the remaining groups as a training data set. Fit a classifier on the training set and evaluate on the test set, retain the test accuracy and summarize after all folds. For each training set, we balance the samples to reach an even split between classes during training, so the classification tasks are entirely balanced. We repeat the procedure for each subject. The mean accuracy from the 5-fold cross validation within each subject is reported as the skill score of the subject. Then we report the mean and standard deviation across subjects. For SVM classifiers, we used the C-support vector classification from Scikit-learn package, with an RBF kernel, regularization=10, and gamma set to be 'scale',$1/(n_{features} \times X.var())$ .

**Stimuli classification comparison across subjects**    Since we perform classification within each subject, we further look into how our model compare to the other methods for each subject. We present the results on the Sherlock early visual ROI and show that higher classification accuracy was achieved with MRMD-AE manifold embedding in 13 out of 16 subjects comparing to the next best method that is a vanilla AE. Moreover, MRMD-AE outperformed individual MR-AE (third-best) in all subjects. Although there are not enough samples to quantify subject-level statistics, we show MRME-AE achieves better performance in almost all subjects( 2).

**Intrinsic dimensions of fMRI data**    We analyzed the intrinsic dimensions of fMRI data by examining the spectral entropy of PHATE diffusion matrix of fMRI data. We refer the readers to Moon

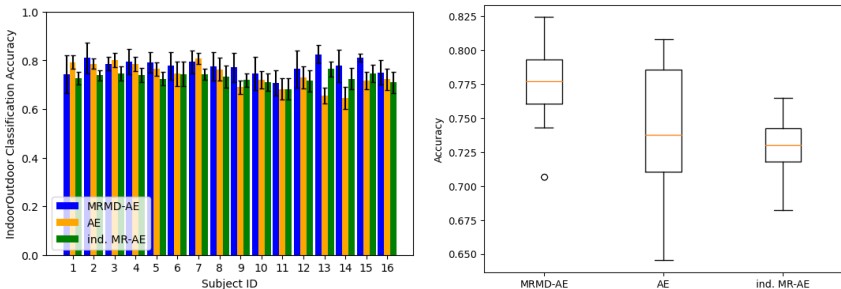

Appendix Figure 2: Classification accuracy comparison across subjects. Classification by indooroutdoor label on Sherlock early visual ROI using the top 3 performing models in Table 1. **Left**: The mean accuracy from the 5-fold cross-validation within each subject was reported. Error bar was standard deviation from the 5-fold cross-validation. MRMD-AE outperforms the next best method, a vanilla AE, in 13/16 subjects and outperforms the third best method, individual MR-AE, in all subjects. **Right**: Box plot of the same set of accuracy values show better performance of MRMD-AE compared to the other methods.

et al. (2019) for how PHATE computes the diffusion operator and the relationship between diffusion operator to the data geometry. For the purpose of current analysis, one can consider the diffusion operator to represent the ambient data geometry and the spectral entropy analysis via Von Neumann Entropy similar to the analysis of principle components that account for most variation in PCA. We report the intrinsic data dimensions of various ROIs of the Sherlock and Forrest fMRI datasets that we use in the experiments of this paper in Appendix Table 1.

| Dataset | Sherlock | | | Forrest | |
|---|---|---|---|---|---|
| ROI | early-visual | pmc | early-auditory | late-visual | early-visual |
| Intrinsic Dimension | 30 | 29 | 41 | 13 | 15 |

Appendix Table 1: fMRI ROI data intrinsic dimensions

