# OpenReview forum: "Learning shared neural manifolds from multi-subject FMRI data"
_ICLR.cc/2022/Conference — ICLR 2022 Submitted_

### Official Review · Reviewer_h5JS · 2021-10-26

**Correctness:** 3
**Technical Novelty And Significance:** 3
**Empirical Novelty And Significance:** 2
**Recommendation:** 5
**Confidence:** 3

**Main Review:**

## Strengths
The problem tackled by the paper is interesting, and the proposed model is original. The model is sound, conceptually interesting, and reasonably described. It is based on recent techniques that are appropriately described and cited. The paper is clear and well structured.

## Weaknesses
1. It is not super clear why the proposed architecture uses a shared encoder and a separate decoder per subject. This choice seems central to the proposed method and should be better explained.
2. It is also not clear which decoder was used to predict on held-out subjects. It seems like a decoder must be trained at some point on the held-out subjects, which might undermine the integrity of the cross-subject experiment.
3. It is not clear how hyper-parameters lambda and mu were set, with a reasonable risk of overfitting the proposed benchmarks. The authors should explain how the hyper-parameters were chosen.
4. "MRMD-AE achieves lower or matching MSE than the landmark approach" does not seem entirely supported by the data. The landmark approach is actually below MRMD-AE at 70% of train set, and the difference does not seem relatively smaller than the difference at other percentages.
5. Tables with numbers are very non-intuitive, and might even be considered a ploy to hide small differences in performances between methods. The authors should use graphs, and move the Tables to the appendix if still considered necessary. If still using tables, the authors should remove non-significant digits (6 digits are not necessary when the standard deviation is at 5e-2.
6. Figure 4: when using barplots, the y-axis should start at 0. Indeed, barplots use our intuitive perception of areas to convey the difference of values, but this perception is biased if the bar does not start at zero, and the difference of values is artificially increased.
7. "We will provide a link to an anonymous repository to the reviewers and area chairs", yet the link was not provided.
8. "fMRI data is now being collected from thousands of subjects" should be tempered by the fact the current study is limited to 16 subjects.

### Unclear sentences
- The "Manifold Regularization Penalty" section is quite hard to follow. The authors might improve it by using shorter sentences.
- What is a "soft" regularization ?
- What is a "batch effect" ? and a "strong subject batch effect" ?
- What is a "individual-subject manifolds subject" ?

### Typos
- [Moon 2019a] and [Moon 2019b] are identical.
- "demonstraging"
- "that that"
- "Reduduction"
- "We therefore compare" (no dot)
- "TRs" is not defined
- Fontsize in Figure 2 is too small.
- "An. SVM"

**Summary Of The Paper:**

The paper proposes a model to learn a low-dimensional representation of fMRI data over multiple subjects of the same experiment. The model is built as an auto-encoder, with an encoder shared across subjects, and a separate decoder per subject. The model is regularized so that the first layer of each decoder gives a representation close to a pre-computed manifold embedding. An optional regularization also constrains the shared representation to be similar across subjects.
The paper then proposes a series of experiments to demonstrate the benefits of the learn representation. The experiments consider the tasks of (a) projecting new test samples to the learned manifold, (b) classify some stimulus features from the embedding (decoding task), and (c) predict brain recordings of a new subject.

**Summary Of The Review:**

My recommendation (reject) is based on
- a missing justification of the architecture design
- a number of missing important details about the fitting procedure
- a poor quality in the reporting of results

---

> ### Author Response · Authors · 2021-11-18
> **Response to Reviewer h5JS**
>
> Thank you for the detailed comments which help us to considerably improve our paper.
>
> On the single encoder multiple decoder architecture: In section 4.2 (page 4 under Figure 1)  we introduced the architecture and explained the rationale, which we repeat here. The rationale for using a common encoder is for the model to learn a common fMRI language from which subject-specific responses can be derived. While the fMRI data exhibits a strong batch effect between subjects, the common encoder builds invariance in the latent space to individual noise in the encoded space. The individual decoders enable incorporating subject-dependent interpretation of the latent information from the common encoder. Thus this framework pushes individual noise into the decoding space. In particular, the decoders can learn to reconstruct the subject-specific signals and recreate batch effects to reconstruct data. The flexibility provided by individual decoders allows the single common encoder to only retain input-dependent variability between subject responses that is sufficient for the decoders to reconstruct the measurements. We will make sure to clarify this even further in the revised text, since as pointed out by the reviewer, it is a central part of our approach.
>
> In the results section, we show that with the subject-specific decoders, we achieve low reconstruction error and can learn manifold embedding for all subjects. We also show that we learn an aligned shared latent space by incorporating the alignment penalty.
>
> ‘Subject batch effect’: fMRI data contains high levels of noise, including scanner noise, physiological factors such as a subject’s blood pressure, head motion and respiration, and individual differences in anatomical and functional topographies in the brain, etc. Moreover, even after accounting for the idiosyncrasies in noise, neural responses evoked by the same stimulus can vary greatly between different subjects, introducing the subject batch effect. We also have a paragraph “Cross-subject batch effects in fMRI” in the preliminaries discussing the problem in more detail.
>
> Due to the cross-subject batch effect, we can only apply the manifold learning algorithm PHATE to each subject’s fMRI data and learn individual manifolds.
>
> On handling the held-out subject: First, our MRMD-AE model employs a single encoder for all subjects which is the central novel part of our approach. We can project any new subject’s data onto the shared latent space and proceed with the low-dimensional embedding of the new data. The idea here is that with end-to-end training of this single-encoder-to-multiple-decoder architecture, the invariant features of multiple subjects used for training will be extracted, and allow the model to generalize to new time points and new subjects. Then we can pass the data through any of the existing decoders and perform the translation task, which projects the new subject’s data onto another subject’s manifold.
>
> On hyper-parameter selection: We did a systematic search for hyper-parameters which is the common approach in the field. The $\lambda$ and $\mu$ used in the experiments are 0.01 or 0.1. The same set of values are used for all subjects for any single task. We will make sure to clearly indicate this in our updated version.
>
> On the performance comparison between landmark approach and our model at 70% train set: Using 70% of data for training, the gap is very small between the MSE of the landmark approach and MRMD-AE (0.00010 vs. 0.00017). However, the stimulus classification accuracy of the landmark-interpolated embeddings remains much lower than that from MRMD-AE (62% vs. 85%). This suggests that although the landmark approach places unseen data on the manifold close to the ground truth, it fails to capture the geometry of the whole dataset and fails to generalize a meaningful manifold to new data. Therefore, MRMD-AE is better at generalizing to new time points.
>
> We apologize for the typos and appreciate the reviewer’s constructive suggestion on improving the presentation. We are making the changes in the updated version.
>
> Here is the anonymous link to our code: https://drive.google.com/drive/folders/1oMrZgZzo-wbZ6Kp_SsIlGG_7-_pxA8tK?usp=sharing

---

> > ### Comment · Reviewer_h5JS · 2021-11-19
> > **Thanks for the detailed response**
> >
> > (Please re-use the numbers associated to each question, otherwise it is hard to navigate your answers, especially if your answers do not follow the same order. Associating each paragraph of your answer to the original questions is not easy.)
> >
> > For hyper-parameter selection, it is still not clear what criterion was used for the selection. In particular, performing the selection on the same dataset that was used for model evaluation would lead to overfitting. This possibility cannot be ruled out given the details provided.
> >
> > Other responses are reasonable, although the paper would require a significant effort to clarify the explanations, and to improve the presentation of the results.
> >
> > (Minor:
> > "Subject batch effect" is a weird terminology to talk about individual differences. "batch effect" makes it sound like it correspond to differences between recording session or between studies, and not differences between subjects.)

---

> > > ### Author Response · Authors · 2021-11-28
> > > **Thank you and further responses**
> > >
> > > Our classification experiment consists of two parts. First, we get the embedding of held-out time points for each subject using the same trained MRMD-AE model. When splitting the timepoints into training and testing sets, we followed the approach in literature (Chen et al. 2015, Chen et al. 2016, Busch 2021) and split the time points into two halves, one for training, the other for testing.  We set the hyperparameters and fit the model on the training sets, then applied the trained model (with the same hyperparameter setting) on the testing sets of all subjects to get the embeddings. As we used the same hyperparameters for all subjects, it is unlikely we overfit to any specific subjects.
> > > In the second part of the experiment, we performed classification on the test embedding of each subject using an SVC (the same setting for all subjects and all classification targets) via 5-fold cross validation. We used the same embedding from each subject (acquired with the same MRMD-AE parameters) for different classification labels (indoor-outdoor,  music-present, etc.), and showed higher accuracies across tasks. Also in training the SVC, we balanced the training samples to reach an even split between classes during training, so the classification tasks are entirely balanced. The risk of our MRMD-AE embedding overfitting to any specific signals is also low.
> > >
> > > We hope this clarifies the hyper parameter selection question. We believe we have addressed all the major concerns of the reviewer. We have also updated the manuscript reflecting most if not all discussions prior to the submission update deadline. Thank you for the helpful comments!

---

### Official Review · Reviewer_nPEU · 2021-11-01

**Correctness:** 2
**Technical Novelty And Significance:** 3
**Empirical Novelty And Significance:** 3
**Recommendation:** 6
**Confidence:** 4

**Main Review:**

# Clear and easy to understand introduction and motivation
The opening sections of the paper were clear and straightforward in motivating the functional alignment problem, manifold-based dimensionality reduction, and generating an approximate manifold embedding operator using a neural net. Likewise, the description of the problem setup, architecture choices and losses builds on itself in a way that was clear and easy to understand. I have only some minor suggestions for improvement here:

[1a] As far as I can tell, the stimulus set $X$ and its elements $X_k$ is only used to say that the encoder maps the fMRI activation map to some representation of the stimulus, but $f'()$ is never defined explicitly. If so, it seems like unnecessary "mathiness" -- the notation is not buying additional precision or insight outside of saying this same thing in words.

[1b] Expression 1 could color-code or otherwise annotate the distinct loss terms (reconstruction, latent closeness penalty, manifold regularization).

# Weak evaluation
However, the empirical evaluation is less well-motivated and weaker overall. A strong empirical evaluation would show [2a] unambiguously state-of-the-art performance relative to [2b] strong baselines, and [2c] evaluations or [2d] ablations that target the core contributions. I detail each in turn:

[2a] Performance benefits seem modest at best. Some very rough statistics: with 15-16 subjects, the confidence interval should be roughly equal to half the width of the SD on each side (since the z- or t-multiplier for the conventional $\alpha=0.05$ is roughly 2 and $\sqrt{15}$ is roughly 4). This means that (very roughly) differences between the best and second-best performing models smaller than the SD may not be statistically reliable. MRMD-AE is rarely convincingly above the next-best model by even a single SD. These statistics are, of course, rough: variability may be much smaller across folds than across subjects, but proper analysis (likely some sort of repeated-measures model) may be needed to tease apart this variability and establish the strength of MRMD-AE more convincingly.

[2b] Strong baselines are missing. First, I'm not sure I understand why SRM is missing from the classification baselines -- it should be applicable here, right? Second, all the baselines are either linear (PCA, SRM) or single-subject (PHATE, AE), and there is no nonlinear functional alignment baseline, even though they exist in the literature (for example: Deep Generalized CCA, Benton et al. ICLR 2017; Deep Hyperalignment, Yousefnezhad & Zhang NeurIPS 2017; Kernel Hyperalignment, NeurIPS 2012 -- and this is only a brief survey). More advanced linear methods likewise exist  (e.g. RSRM, Turek et al. ICASSP 2018; MN-SRM, and DP-SRM, Shvartsman et al. AISTATS 2018). Comparing to all of these methods is not critical but it's hard to get a sense of whether the new method truly exceeds the state of the art without at least a few stronger functional alignment baselines.

[2c] As I see it, the key contributions here are the one-in-many-out architecture, and the additional manifold and latent space closeness regularization terms. A full set of evaluations would probe each in turn, showing how the present contribution performs best in out-of-sample performance of some reconstruction metric, some latent space closeness metric, and some manifold alignment metric. The paper reports untrained cross-subject translation (which is a sort of out-of-sample reconstruction metric), and the future-point-placement metric (which is sort of a latent space closeness metric). The benefit of the manifold regularization penalty is seen in the comparison to MR-AE. Importantly, the overarching narrative present in the front matter of the paper is much weaker here, and the link made above between the contributions and the evaluations is not made explicit.

[2d] The ablation study is not particularly informative about the contributions of components outside of the translation penalty (since there is no model with translation but no alignment penalty, and no model with translation but no manifold regularization). To be clear: I don't think ablations studies are critical to this paper -- but the specific ablation study in the submission could be stronger.

# Minor comments and typos
- In the discussion, "For many studies, fMRI data is now being collected from thousands of subjects" is optimistic. I only know of the HCP and IMAGEN projects, both of which are massive multi-institution efforts. Most fMRI studies are still quite small (tens of subjects at most).
- It would be nice (but not critical) to see some empirical demonstration of the benefit of splitting g() into g'() and g''(), and verification or citations of the claim that applying PHATE to the group data fails.
- The motivation for the time-splitting experiment is a bit odd: in my experience pilot fMRI collections are still full length collections from different subjects, not short collections from the same subjects.
- Figure 2 should identify the dataset used (I think it is Sherlock?).
- I wonder if part of the reason that MRMD-AE fails to perform well on the localizer task is that it overfits to movie data, i.e. it is capturing movie-specific variability that is not present in the localizer data.
- Section 3 first paragraph "train data" -> training data
- Page 5 under "manifold regularization penalty" paragraph -- "individual-subject manifolds subject", second "subject" seems extraneous.
- Last line of section 4: subjects fmri activities -> subjects' fmri activities.
- First sentence on p7 Stimuli Classification section: "unseen times series. fMRI data" -> "unseen fMRI data" or "unseen time series" or "unseen time series of fMRI data" etc.

In sum, I think issues [2a] and [2b] are most critical and keep the paper from clearing the bar for ICLR. Addressing [2c] and to a lesser extent [2d] would also strengthen the paper, but would not be sufficient if [2a] and [2b] were not addressed.

EDITED: I am adjusting my rating to a 6 based on additional information provided about the raw classification scores, but I'm still concerned about the baselines chosen.

**Summary Of The Paper:**

The submission proposes a new model for functional alignment of fMRI datasets from multiple subjects. It combines a one-in-many-out autoencoder with two regularization loss terms (one inspired by GRAE) to develop a model that can encode every subject's data to a shared latent space, from which each subject's data is then decoded by a separate decoder. Experiments are provided that demonstrate benefits to some downstream tasks.


**Summary Of The Review:**

I think this is a well-motivated approach and the early parts of the paper are a pleasure to read: it was easy to understand what each piece of the contribution is meant to do, and its motivation and relation to past work. When it comes to the empirical evaluation, however, the paper runs into a common problem with noisy fMRI data, which is that it's hard to convincingly beat linear methods, and it then overstates (to my read) the strength of its empirical contributions. The performance gains are modest and nonlinear functional alignment methods (e.g. from the hyperalignment family such as KHA/DHA) and more advanced linear methods (such as RSRM and MN-SRM) are not included in the comparison. The structure of the latter portions of the paper suffers as well: while the specific modeling choices are well-motivated, the experiments do not fully probe the novel contributions. As a result, while I think this line overall is worth pursuing, I would like to see stronger empirical results for this work to be a successful ICLR submission.

---

> ### Author Response · Authors · 2021-11-18
> **Response to Reviewer nPEU (Part 2)**
>
> [2b] The classification analysis is a within-subject classifier on the individual subject manifold. However, SRM finds common signals shared between subjects and evaluations of these metrics are performed on the common space representations.  Therefore running the classification on the shared SRM space would not be an equal comparison.
> Our baselines were the most common, dimensionality-reduced functional alignment methods in the field used for naturalistic stimuli. Methods such as CCA, deep hyperalignment, and kernel hyperalignment, though nonlinear, do not learn reduced-dimensionality common spaces and are not commonly used in the field. We will make a point to discuss their potential but kindly ask the reviewer to acknowledge that a pragmatic choice of baseline comparisons, as we believe we have done here, is required given the limited scope and length of an ICLR paper.
>
> [2c] You are right. We used the untrained cross-subject translation to show out-of-sample reconstruction, and the future-point-placement to show latent space closeness.
>
> [2d] We show that adding the alignment penalty achieves better translation as it allows for learning a more aligned shared latent space, which is a desirable feature (Table 3, Figure 3). Then adding the translation penalty further improves translation (Figure 4). We will further emphasize this point in the text to avoid confusion.
>
> On the two parts of each decoder. As shown in Figure 1, the first layer of the decoder is what we refer as $g_j’$, we applied the manifold regularization penalty as in the equation at the end of the subsection “Manifold Regularization Penalty” to match the embedding after this first layer of decoders ($g_j’$)  with the PHATE embedding. The PHATE embeddings were constructed for each subject, thus are subject-dependent. In the first subsection of experimental results ‘Place Future fMRI Time Points on Learned Manifold’, we showed that the MRMD-AE generated manifold embedding closely match the ground-truth subject-specific PHATE embedding (Figure 1), confirming that the $g_j’$ layer introduces the subject dependent batch effect back, approximates the per-subject PHATE manifold and allows for the following reconstruction by the rest of the decoder layers. The rest of the decoder layers $g_j^{\prime\prime}$ reconstruct the fMRI data from the manifold embedding. For reconstruction, we reported low reconstruction errors in Table 3. We apologize for the confusion, but our intention was to make it more clear to the readers where we applied the manifold regularization and separate the shared latent space out of the single encoder from the subject-dependent manifold latent space after the first layer of the decoders.
>
> We will correct the typos and improve the presentation as suggested in [1a],[1b], and minor comments. Thank you! We hope that these responses address your main concerns and that you will consider revising your score as a result.
>
> Here is the anonymous link to our code: https://drive.google.com/drive/folders/1oMrZgZzo-wbZ6Kp_SsIlGG_7-_pxA8tK?usp=sharing

---

> ### Author Response · Authors · 2021-11-18
> **Response to Reviewer nPEU (Part 1)**
>
> We thank the reviewer for the thoughtful review and suggestions to help us improve our paper.
>
> [2a] We apologize that the reported statistics we gave were confusing. We will try to clarify this. We used the data of 16 subjects, and within each subject we performed 5-fold cross-validation. We had given a variance and standard deviation of performance across subjects. However, we realize that these are too few subjects on which to formulate any meaningful statistics.
>
> In response to this, we note here that the MDRM-AE creates embeddings of voxel activation patterns at points in time and not the entire subject. The points of embedding are activation patterns corresponding to the timepoints of measurement at which there are 1976 for the Sherlock data. Within this we have classification accuracy results on the timepoints. These can be averaged across K-folds of validation or randomized based on initialization of the network, which we have added to the manuscript in the supplementary materials.  However, statistics cannot be meaningfully formed at a “per subject” level. We only have 16 subjects, within which we show that we can classify voxel-activation patterns correctly by stimuli labels accurately in all cases. On the Sherlock early visual ROI, the classification accuracy of our model is higher than the next best method, which is a vanilla AE, in 13/16 of the subjects but these are not enough samples to quantify subject-level statistics. Our model outperforms the third-best method, which is the individual MR-AE in all subjects. We include the classification accuracies for each subject in the following table.
>
> | Subject ID | MRMD-AE |   AE  | ind. MR-AE |
> | ---------- | ------- | ----- | ---------- |
> | 1          | 0.743   | **0.792** | 0.727      |
> | 2          | **0.809**   | 0.785 | 0.738      |
> | 3          | 0.785   | **0.801** | 0.745      |
> | 4          | **0.793**   | 0.786 | 0.739      |
> | 5          | **0.792**   | 0.765 | 0.724      |
> | 6          | **0.777**   | 0.745 | 0.744      |
> | 7          | 0.793   | **0.808** | 0.742      |
> | 8          | **0.774**   | 0.764 | 0.734      |
> | 9          | **0.771**   | 0.690 | 0.719      |
> | 10         | **0.745**   | 0.720 | 0.710      |
> | 11         | **0.707**   | 0.681 | 0.682      |
> | 12         | **0.764**   | 0.730 | 0.716      |
> | 13         | **0.824**   | 0.655 | 0.765      |
> | 14         | **0.777**   | 0.645 | 0.724      |
> | 15         | **0.810**   | 0.717 | 0.745      |
> | 16         | **0.750**   | 0.722 | 0.709      |

---

### Official Review · Reviewer_Rq5W · 2021-11-01

**Correctness:** 3
**Technical Novelty And Significance:** 3
**Empirical Novelty And Significance:** 3
**Recommendation:** 8
**Confidence:** 4

**Main Review:**

The paper makes an interesting and well-motivated contribution on a field typically overlooked in ICLR. Although some aspects of this paper exist in previous literature, to the best of my knowledge the ideas are novel and based on well-thought motivations stemming from specific challenges in the field of fMRI data. The empirical results - although in my opinion not sufficiently thorough - give a good indication that this new approach can be successfully applied on a multitude of tasks, improving previously used techniques. Given these reasons, I recommend acceptance of this work. However, my recommendation is only marginally above the acceptance threshold because the paper contains crucial flaws, mainly (1) a lack of sufficient ablation analysis and (2) several portions of the paper requiring further clarification. In this review I’ll first list what I think are the major weaknesses of this work, leaving some minor remarks and suggestions after.

# Major weaknesses
1. The paper mentions a “translation” task several times throughout the paper, but it never explains what it specifically consists of. Given this seems to be the task where the proposed architecture is the least strong (from Table 3), this is an important part to clarify.
2. In the third paragraph of the Introduction, the paper mentions that typical autoencoders “often just spread out points for ease of decoding”. Many of the variational autoencoders developed from vanilla autoencoders actually try to solve this issue with AEs, by making the latent spaces more meaningful and smooth. Given how a direct development from AEs seems to directly tackle many of the issues which motivated the paper, I do not understand why the paper didn’t use VAEs as another baseline for comparison, and I find this to be an important weakness in the paper.
3. The paper mentions in the “Related Work” section that it used a similar multi-objective NN approach like in SAUCIE; however, it is not clear what are the differences to this previous method which would grant novelty.
4. Section 4.1 introduces a concept of common stimuli X for all subjects. Furthermore, this concept is used to show the properties that functions “f” and “g” should have, specifically in (b), (c ), and (d). However, this concept (ie., X) is not explored anywhere else in the paper, nor these properties are validated in the experiments. I find this to be a major point of confusion for the reader.
5. Section 4.1 mentions “intrinsic dimensions of the data” but I think this concept is not explored nor explained (including in the appendix table referenced).
6. A big point of the paper is how PHATE seems to correct for strong subject batch effects existent in fMRI, and how this is important in creating a subject-independent manifold. Although I think Figure 3 supported the claim of a subject-independent manifold, I don’t see how batch effects were accounted for at all; usually they relate to confounds like sex, ethnics, BMI, and others. In any case, why would batch effects be a problem in this case, as it is well-known how they can influence fMRI timeseries and therefore could be important for a rich low-dimensional manifold?
7. A strong weakness of this paper is the lack of a more detailed ablation analysis on the several components introduced in the architecture. For example, the paper splits each decoder into two (Subsection “Manifold Regularization Penalty”) and tries to motivate this split into g’ and g’’ using very loose intuitions which I’m not even sure I agree with; an ablation analysis would further complement the paper to see how much influence this has in practice.
8. Many details are missing in the paper to properly understand how the classification tasks are conducted. How does the paper decide where to “cut” the timeseries in order to conduct the SVM classifier and how is the 5-fold CV conducted/divided?  What hyperparameters were used for the SVM classifier? Given the datasets seem to have a label for each input, what is the reasoning for using a classification over many timepoints?
9. The paper claims “We show that our model achieves best translation performance on various brain ROIs in at least one metric”. I do not understand where this is shown, given no figure or table is referenced in this paragraph, nor the concept of ROIs seem to be explored here.

# Minor comments and suggestions
1. I want to show my appreciation for the reproducibility statement, but as a suggestion, an anonymised code would have made this submission stronger and potentially clarify some of the questions I am asking in this review.
2. Some acronyms appear in the text without being defined or specified with a reference beforehand, for instance BCI, BOLD, PCA and SVD. Although it’s reasonably fair to assume the majority of the ICLR community will know some of these terms, defining them it’s always better for preciseness, clarity, and avoidance of possible confusions for people in other fields reading this paper. I think for a computer science conference like this, the definition of BOLD signal should go beyond just spelling the acronym, as probably many people in the conference do not know what this means (e.g. in the Preliminaries section).
3. In the fields of neuroscience, a standard dataset for any fMRI study is the Human Connectome Project (HCP) or, more recently, the big UK Biobank. Were these more widely-used datasets not used for some specific reasons? I’m aware that the UK Biobank requires a long application process, but all HCP data are publicly available.
4. As a suggestion for future work, I’d say it would be interesting to use resting-state fMRI which is typically quite noisier than the task-specific fMRI used in this work. I believe it would be interesting to see whether the experiments on the manifold extension task would still work here.
5. How imbalanced are the classification tasks? This is important to understand whether accuracy is a good-enough metric for this task.
6. The paper claims that tSNE and UMAP “typically do not preserve overarching trends or relations between data regions”. For a complete related work section, the paper should probably mention that this is a contested statement and not so widely accepted as the paper claims. For example, some use UMAP for classification tasks (https://umap-learn.readthedocs.io/en/latest/supervised.html), and recent works explored preservation of global structure on real-world data, for instance: (1) https://www.nature.com/articles/nbt.4314, (2) https://www.nature.com/articles/s41540-021-00186-6, (3) https://www.nature.com/articles/s41467-020-15351-4.
7. Footnote (1) is confusing. How can one time point in one region of interest be a vector of length equal to the number of voxels? When the functional images are aligned to a common template, it is common to extract a single averaged timeseries per region of interest, otherwise why is this alignment done, just as a preprocessing step? Indeed, from the definition it seems that each subject only has in fact one single timeseries, as Y only varies across j (subjects) and k (timepoints) as subscripts. This needs to be further clarified.

# Minor comments to improve clarity and readability
1. The paper uses a metric named MDS without explaining what it actually means. I don’t believe this is a metric well-known by the community.
2. “ROI” is not defined in the paper even though it is used in a few places.
3. In Section 4.1, point (c ) starts with “They”; please define what this actually corresponds to.
4. In “Embedding Alignment Penalty” (in Section 4.2) there’s a typo in “common decoder (and individual encoders)” (should be the other way around)
5. “TRs” is not defined in Section 5.
6. How did the paper arrive at this set of hyperparameters? Manual choice? Hyperparameter sweep?
7. Typo in “bottle neck” (needs to remove the space) (Section 5, “Autoencoder Hyperparameters”)
8. Figure 2: yaxis should be only MSE (without the “vs” part)
9. Figure 2: I find the MSE values extremely small, and the confidence intervals overlapping a lot. Therefore, I do not agree with the paper’s claims that MRMD-AE achieves significant results here. I believe the paper needs to tone down their conclusions in this subsection or explain how this small value (i.e., below 0.001) can be so significant.
10. Typo: “An. SVM” (begin of page 7)
11. Typo: “unseen times series. fMRI” (Section 5, “Stimuli Classification”).
12. The paper uses a metric called “Earth Mover Distance” to support some of their claims (eg in Figure 3). However, this metric is not explained.
13. In one paragraph in Section 5 it is written “SVM classifier” and 2 paragraphs later it is instead “A support vector classifier (SVC)”. Are these the same or did something change?
14. Tables: I believe it would improve readability to reduce the number of decimal points. 2 decimal points in accuracy and 3 in standard deviation would probably suffice.
15. Tables: What is the difference between “AE” and “Individual MR-AE”? If it is the usage of PHATE on MR-AE but not on an equivalent autoencoder with shared encoders and decoders, then please say in the text (and indeed this is a good ablation).
16. Table 3 is not referenced in the text.
17. Figure 3: Are the PHATE embeddings already in 2D space, or did the paper have to apply some technique like UMAP or tSNE for visualisation?


**Summary Of The Paper:**

The paper proposes a new neural architecture named MRMD-AE that can be applied on noisy fMRI data in different tasks. Subject-specific decoders are used to more directly recover individual signals, while the encoder is shared across every subject under the assumption that every person will share common low-dimensional features for the same stimuli. A key component of this architecture is the usage of a regularisation term named PHATE (previously introduced in the literature) which allows the latent space to not be split into individual embeddings and be extendable for unseen data. The paper empirically shows how this architecture improves metrics on classification tasks when compared to previously used techniques.

**Summary Of The Review:**

Overall I think this is a very interesting and useful idea in the convergence of the fields of neuroimaging and learning representation. However, the lack of important experiments and explanations make me recommend only marginally above the acceptance threshold. I believe my questions can be mostly (if not all) addressed during a rebuttal process, after which I’d be happy to change my score as I think overall this should be shared among the ICLR community.

------------------ EDITED AFTER REBUTTAL PERIOD:
I followed the authors answers to my questions and other reviewer's questions. Although the authors didn't address all my questions (for example my comment titled "Further discussion") nor they uploaded a new pdf version to check corrections, I believe my main concerns were clarified. Assuming the authors introduce the written clarifications in the paper, I'm changing my review recommendation in two points:
- Changing "Correctness" from 2 to 3.
- Changing my "Recommendation" from 6 to 8.

---

> ### Author Response · Authors · 2021-11-18
> **Response to Reviewer Rq5W (part 3)**
>
> On the datasets we present here: We specifically use movie watching datasets because there is a growing trend in cognitive neuroscience research toward using naturalistic movie stimuli as they more closely resemble our day-to-day, real-world experience compared to simple task paradigms or rest. Movies sample a larger range of cognitive processes as subjects infer the social interactions among characters, make predictions about events, experience a rich audiovisual environment, and engage memory processing on multiple timescales. Further, these movie-watching datasets have thousands of timepoints per subject. The HCP includes task based data from an array of typical block-design tasks (N-back, Posner, gambling, etc) that aim to engage specific cognitive mechanisms one at a time, rather than simultaneously. The UK Biobank, while it has a huge number of subjects, has approximately 4 minutes of task data per subject. In cognitive neuroscience research, the Sherlock and StudyForrest datasets are among the most popular datasets using naturalistic stimuli. In the future, it would be interesting to see how this extends to resting state data, though it would be harder to validate the model’s alignment performance without a common stimulus and classification labels.
>
> On improving presentation. We apologize for the typos and undefined acronyms. Thank you for all your suggestions and for the close inspection of our paper. We highly appreciate it. We are incorporating all the suggestions in our updated version.
>
> We apologize for the delay. Here is the anonymous link to our code: https://drive.google.com/drive/folders/1oMrZgZzo-wbZ6Kp_SsIlGG_7-_pxA8tK?usp=sharing

---

> ### Author Response · Authors · 2021-11-18
> **Response to Reviewer Rq5W (part 2)**
>
> On intrinsic dimensions of data and PHATE: We explained the intrinsic dimension analysis in a paragraph before the Appendix Table 1. We analyzed the intrinsic dimensions of fMRI data by examining the spectral entropy of PHATE diffusion matrix of fMRI data.  We refer the readers to the PHATE paper for how PHATE computes the diffusion operator and the relationship between the diffusion operator to the data geometry. For the purpose of the current analysis, one can consider the diffusion operator to represent the ambient data geometry and the spectral entropy analysis via Von Neumann Entropy similar to the analysis of principal components that account for most variation in PCA. Thus the number of intrinsic dimensions via spectral entropy analysis is comparable to the number of PC that account for 99% explained variance. We would be happy to discuss more details of the PHATE diffusion process and analyses, which is an interesting but separate topic from this paper.  PHATE learns the data manifold that preserves local and global data structure and 2-3 dimensional embeddings ready for visualization. One issue of PHATE is that it cannot reliably extend to new data, which our approach addresses.
>
> On cross-subject batch effect: fMRI data contains high levels of noise, including scanner noise, physiological factors such as a subject’s blood pressure, head motion and respiration, and individual differences in anatomical and functional topographies in the brain, etc. Moreover, even after accounting for the idiosyncrasies in noise, neural responses evoked by the same stimulus can vary greatly between different subjects, introducing the subject batch effect. We also have a paragraph “Cross-subject batch effects in fMRI” in the preliminaries discussing the problem in more detail.
>
> On the two parts of each decoder: As shown in Figure 1, the first layer of the decoder is what we refer to as $g_j'$, we applied the manifold regularization penalty as in the equation at the end of the subsection “Manifold Regularization Penalty” to match the embedding after this first layer of decoders ($g_j’$)  with the PHATE embedding. The PHATE embeddings were constructed for each subject, thus are subject-dependent. In the first subsection of experimental results ‘Place Future fMRI Time Points on Learned Manifold’, we showed that the MRMD-AE generated manifold embedding closely match the ground-truth subject-specific PHATE embedding (Figure 1), confirming that the $g_j’$ layer introduces the subject dependent batch effect back, approximates the per-subject PHATE manifold and allows for the following reconstruction by the rest of the decoder layers. The rest of the decoder layers $g_j^{\prime\prime}$ reconstructs the fMRI data from the manifold embedding. For reconstruction, we reported low reconstruction errors in Table 3. We apologize for the confusion, but our intention was to make it more clear to the readers where we applied the manifold regularization and separate the shared latent space out of the single encoder from the subject-dependent manifold latent space after the first layer of the decoders.
>
> On the translation task results: We apologize for the missing reference to Table 3 where we showed the cross-subject Translation Comparison when MRMD-AE is not trained for translation.The translation across subjects $\hat{Y}_{i->j}=g_j(f(Y_i)), i\neq j$ is compared with the target subject $Y_j$. MRMD-AE achieves lower MSE and higher correlation (at least one of the two) across  ROIs. ROI refers to Region of Interest, e.g. early-visual, early-auditory, pmc, etc.
>
> On tSNE and UMAP: We found that tSNE and UMAP tend to create clusters even when they do not exist, and shatter the data trajectory we expect to observe from biological data and fMRI time series. Moon et al. 2019 and Kuchroo et al. 2021 discussed this effect in detail on multiple datasets. Duque et al. also discussed in the GRAE paper that UMAP tears the overall structure of the manifold on the Swiss Roll and a single cell cytometry dataset despite their known continuous natures. We believe that PHATE is better suited for application in fMRI data considering the continuous nature of fMRI time series data.
>
> On fMRI data used in experiments referring to footnote 1: When preprocessing raw fMRI data, it is aligned to a common template (MNI 3mm template here) so that all subjects can be evaluated in the same space. From there, as is commonly done in multivariate pattern analysis- type fMRI literature since 2001, we extract the data for each voxel within a region of interest and treat it as a matrix of [number_of_timepoints, number_of_voxels_in_region]. No averaging is performed here - each voxel is treated as a feature, and each timepoint is treated as a sample. For example, the early-auditory ROI contains 1018 voxels, so the data is 1018-dimensional. The Sherlock dataset has 1976  time points, so each time point is a 1018-dimensional vector.

---

> ### Author Response · Authors · 2021-11-18
> **Response to Reviewer Rq5W (part 1)**
>
> Thank you for the detailed review and constructive suggestions, which help us improve our paper. We appreciate the opportunity to answer your questions and clarify aspects of the paper.
>
> On clarifying the translation task. Our MRMD-AE framework allows integration of data from multiple subjects, thereby enabling translation between subjects. Precisely, given fMRI data from a total of $n$ subjects, we train the model with the $m_{train}$ TRs of all subjects. By training the model with only the $m_{train}$ TRs, we prevent data leakage from the test data of any subjects.  Then for every subject $i$, we translate its test TRs to another subject $j$, $j\neq i $ by $\hat{Y}_{i->j} = g_j(f(Y_i)) $, where $f$ is the shared encoder and $g_j$ is the decoder of subject $j$. Through the process, we project the fMRI data of a new subject to the shared latent space through the encoder, then reconstruct the data in another subject’s ambient space, achieving translation cross subjects. We described the process in the results section but will move it to earlier parts of the paper to facilitate reading.
>
> On VAE. VAEs feature a KL-divergence penalty which creates a contiguous latent space. However, they are not in general smooth with respect to the manifold geometry (i.e., manifold intrinsic distance preservation), which is a major pursuit of our paper. We do not expect the VAE by itself to resolve the issue of cross-subject batch effect in the latent space. In fact, Duque et.al. in the GRAE paper (which uses geometry regularization) discussed that the VAE Gaussian prior, while useful for generative sampling, is inadequate to model the true manifold of real-life (biological) datasets, and is not suitable for general manifold learning. Moreover, Moor et. al. in their paper on Topological Autoencoder found that VAE failed to discover the structure of data and they had to use the topological constraints to identify the structure of data.
>
> On SAUCIE. SAUCIE is an autoencoder-based method that uses regularizations to encourage interpretable latent representations for single cell data, but it is not of the single encoder multi-decoder architecture and does not learn manifold embedding and the loss penalties are not applicable to fMRI data. We creatively chose loss penalties that are specific to our target problem, which is extracting meaningful low-dimensional manifolds and learning aligned shared latent space from fMRI data. The *single encoder multiple decoders* structure of our MRMD-AE framework is also novel and makes learning a shared latent space that corresponds to the common stimuli from multiple subjects possible.
>
> On common stimuli. Indeed the subjects all received common stimuli in our datasets.  For example, In the Sherlock data, fMRI data was collected when all subjects were watching the same episode of BBC series Sherlock. Thus, they were exposed to the same stimuli because the same movie segment was on screen at the same timepoints for every subject. The labels (whether the scene happened indoor or outdoor, if music was present at *each time point of the movie*) of this common movie stimuli are what we used as decoding targets for the classification. By showing higher classification accuracies, we confirm that our model preserves signals of the common stimuli. Moreover, we showed that we learn an aligned shared latent space out of the encoder. This also suggests that the shared latent space is separated from subject-dependent signals and captures the key structure of the common stimuli.
>
> On details of the classification task. The movie labels are for each time step. Consider the Sherlock dataset, where fMRI data was collected when all subjects were watching the same episode of the BBC series Sherlock. The movie and the fMRI data both have 1976 time points. Each time point is annotated with labels (used as target for classification) including whether the scene happened indoor or outdoor, and if music was present at that time point of the movie. Therefore we can treat each fMRI timepoint of a subject as a sample for classification tasks. The input dataset for each subject contains 1976 samples. For 5-fold CV, we proceed following the standard procedures. We split the dataset into 5 groups at random, and for each group we take as a hold out or test dataset and use the remaining groups as a training data set. Fit a model on the training set and evaluate on the test set, retain the test accuracy and summarize after all folds. For each training set, we balance the samples to reach an even split between classes during training, so the classification tasks are entirely balanced. For SVM classifiers, we used the C-support vector classification from Scikit-learn package, with an RBF kernel, regularization=10, and gamma set to be $1 / (n_{features} \times X.\text{var}())$. We are adding the experimental details to the supplementary materials.

---

> ### Comment · Reviewer_Rq5W · 2021-11-20
> **Further discussion**
>
> I thank the authors for the effort in answering my question as well as the other reviewers's questions which I've also read. After reading all the answers I think there are many confusing parts of the paper that are much clearer now - this means that the authors should really make an effort in incorporating many of their answers in the reviewed paper. I look forward to see the revised part, which will probably have to contain some supplementary material sections. I also want to show my appreciation for the anonymous code shared in a drive folder.
>
> I just have a few more questions/comments.
>
> 1. On the topic of the VAE, I want to thank the clarification. I appreciate that the focus of the paper was on the manifold geometry, and therefore I understand why the authors argue that VAE is not suitable. However, in that sense, PCA and AE are even less suitable for a comparison and they are still used in the paper. My argument is that the paper would benefit from a comparison with a method that partially meets some of the motivation in the paper (ie smoothness in the learned embedding) - instead of using PCA/AE which do not meet any of the paper's motivations; therefore, a VAE would probably be a more fair baseline. Despite the literature pointed by the authors on the issues with the VAE's embeddings, I'm not sure whether this is such a widely accepted view given some works like Tybalt (https://github.com/greenelab/tybalt) that are based on VAEs and have produced some biologically interesting results.
> 2. Did I miss something or the authors didn't tackle this comment: "Section 4.1 introduces a concept of common stimuli X for all subjects. Furthermore, this concept is used to show the properties that functions “f” and “g” should have, specifically in (b), (c ), and (d). However, this concept (ie., X) is not explored anywhere else in the paper, nor these properties are validated in the experiments. I find this to be a major point of confusion for the reader.". I'm specifically saying that besides the clarification of the common stimuli that should be in the final reviewed paper, the properties (b), (c), and (d) don't seem to be validated in the experiments.
> 3. I'm not sure the authors addressed my previous comment: "A big point of the paper is how PHATE seems to correct for strong subject batch effects existent in fMRI, and how this is important in creating a subject-independent manifold. Although I think Figure 3 supported the claim of a subject-independent manifold, I don’t see how batch effects were accounted for at all; usually they relate to confounds like sex, ethnics, BMI, and others. In any case, why would batch effects be a problem in this case, as it is well-known how they can influence fMRI timeseries and therefore could be important for a rich low-dimensional manifold?". They mention indeed other well-known batch effects from fMRI data, but I still don't understand how they were accounted for in practice, or how the authors prove that indeed they are correcting for batch effects.
> 4. The authors clarification on g' and g'' make total sense (and I do hope this clarification is included in the reviewed paper). However, in the paper itself the authors motivate this division in another way, specifically "we split each decoder into g_j = g'_j ◦ g''_j, where intuitively we expect the first part g'_j to “implement batch effects” going from the common latent representation to a subject-dependent one that captures well the intrinsic structure of their stimuli responses, while the second part g''_j will then map this intrinsic represention to the observable measurements.". I'm not sure how this is confirmed in the experiments, which comes back to my point that an ablation study would be beneficial in this case. All the results explore only "g", rather than this division into g' and g''. Am I missing something?
> 4.1. While copying your sentence in my previous point (4), I noticed you have another typo in "represention".
>
>
> When answering direct points, it would be useful for the authors to use the same numbers as used by the reviewers to be easier to see what points specifically the authors are answering to (instead of using many paragraphs).

---

> ### Comment · Reviewer_Rq5W · 2021-11-20
> **A recommendation**
>
> I finish with a recommendation after I saw the authors' answers to the other reviewers regarding statistical testing. Please do not justify something as just because it's "common" or "standard" - that is no valid justification at all. Reviewers were asking for some statistical testing because of a concern that I also share with the other reviewers: results seem extremely similar and the confidence intervals overlap a lot, therefore we were trying to ask for something more that could better support the authors' claims. Just like pvalues, only using an averaged metric with standard deviation to compare models, or only using a single metric (ie accuracy) in the tables when it's not clear how imbalanced the datasets are, have their own limitations too. The recent literature criticising p-values was less about p-values themselves, and more about how people use them. From the plots in the paper, it seems very likely that a p-value testing whether the means are different would not return a small p-value like the authors claimed in one of their answers. In any case, I agree with the authors that given only 5 folds, that is not enough for a meaningful statistical test - which again reduces the significance of the work in this very specific case. But I had to make this remark.

---

> > ### Author Response · Authors · 2021-11-21
> > **Regarding statistical significance tests**
> >
> > Thank you for providing this recommendation, and for acknowledging that in this setting statistical testing may not be meaningful compared to other settings.
> >
> > Regarding "common" or "standard" practices in the field: such standards do not arise in a vacuum. In the ML community, the assertion of which method is considered SOTA is nearly always based on scoring methods via mean + std of accuracy (or similar scores taking into account unbalanced classes) on standard benchmarks - and typically one method is considered as outperforming another when the accuracy reported is consistently higher on most benchmarks. This is the case in vision tasks, NLP, graph classification, node classification, etc. - the examples are numerous and easy to find. As such, they reflect the consensus regarding the definition of what threshold is sufficient to consider (within ML, or at least deep learning) one method as outperforming another. Most significance tests anyway would require a common standard for null hypotheses and certainty thresholds to assert whether differences are random or not, so referring to the standards employed in the field is highly relevant, otherwise - should we ignore most of the advances reported in recent years?
> >
> > Now, at this point one can ask why and how did the DL community reach such a wide consensus based on these metrics rather than consider some statistical test (whether p-value, t-test or otherwise). Most statistical tests essentially try to assert whether two (conditional) random samples are coming from the same distribution or not, and set a threshold of confidence on when can we reasonably assert they do in fact come from different distributions. It is not entirely clear here what source of randomness would make sense in the context of neural networks when we consider the distribution of results. One source is from the splits that are used in cross validation, but typically very few folds are tested, and anyway this clearly assumes the train and test distributions are the same, which is unrealistic. This, in turn, is the reason multiple benchmark datasets are often being used, with varying difficulties or ranges of possible results. In this sense, it would be difficult (if not impossible) to come up with a realistic null hypothesis over datasets.
> >
> > More importantly, even if one accounts for the sampling or randomness in the cross validation procedure, neural networks are highly non-convex models, which means that another level of randomness inadvertently comes in from the fact that each test-train split effectively results in a different neural network model. As high capacity models with an extremely high dimensional space for their parametrization, it is clearly unrealistic to expect any given architecture to result in anything close to similar instantiations from different initializations. This is a much more significant source of randomness than anything coming up from the cross validation procedure - using different seeds on exactly the same fold (or split) would result in completely different networks (albeit with the same architecture), i.e., different weights and bias terms. Therefore, to perform any significance test, we would have to consider sufficiently many retrained networks to capture (to some level of confidence) the distribution of local optima for a given architecture. Note that this is a major difference between DL and more traditional convex optimization models with a single global optimum.
> >
> > Due to the dimensionality and nonconvexity of the loss landscape here, it is clear that it would be prohibitively expensive to retrain nontrivial architectures to the extent needed to properly capture the distribution of local optima. Similarly, it would also be intractable to compute any a priori estimate of this distribution (or even the probability of reaching a good optimum to begin with) given all possible instantiations, which vary due to initializations, stochasticity in the SGD process and other factors. Moreover, it is unclear what type of null hypothesis would we even have here, or what underlying distribution of results conditioned on such models. Clearly, there's no reason to believe it will be any of the classic ones (normal, t-distribution, etc.) These difficulties have already been noted nearly 20 years ago (see https://www.jmlr.org/papers/volume5/grandvalet04a/grandvalet04a.pdf) and none of the solutions have been taken up as a standard (e.g., as discussed here: http://www.public.asu.edu/~huanliu/dmml_presentation/T-test.pdf)
> >
> > For these reasons, the standards in the field have settled on asking - if you run architecture A and architecture B several times on several datasets, and several folds within each one, which one empirically seems to end up with a better score (say, accuracy) more often than the other, and choose the architecture to use based on this type of "test" without quantifying the chance of this being due to some randomness in the training/testing process.

---

> ### Comment · Reviewer_Rq5W · 2021-11-28
> **Updating my recommendation**
>
> I followed the authors answers to my questions and other reviewer's questions. Although the authors didn't address all my questions (for example my comment titled "Further discussion") nor they uploaded a new pdf version to check corrections, I believe my main concerns were clarified. Assuming the authors introduce the written clarifications in the paper, I'm changing my review recommendation in two points:
> - Changing "Correctness" from 2 to 3.
> - Changing my "Recommendation" from 6 to 8.

---

> > ### Author Response · Authors · 2021-11-28
> > **Re: Updated recommendation and further discussion**
> >
> > We would like to thank the reviewer again for actively engaging in the discussion and for all the helpful and constructive comments.
> >
> > We continue to answer a few more questions in “Further discussion”. Some of the items we copied from the previous responses.
> >
> > 1. Indeed, VAE is not suitable for manifold learning and we do not expect it to produce smooth latent space with respect to manifold geometry. We appreciate the reviewer’s suggestion that it may be of independent interest to further explore the usage of VAE and its adaptations to investigate biological data and compare it with our results.
> >
> > 2. On common stimuli: Indeed the subjects all received common stimuli in our datasets. For example, In the Sherlock data, fMRI data was collected when all subjects were watching the same episode of BBC series Sherlock. Thus, they were exposed to the same stimuli because the same movie segment was on screen at the same timepoints for every subject. The labels (whether the scene happened indoor or outdoor, if music was present at each time point of the movie) of this common movie stimuli are what we used as decoding targets for the classification. By showing higher classification accuracies, we confirm that our model preserves signals of the common stimuli. We can perform the classification task either on the shared embedding space ,$f(Y_{j,k})$, or the manifold embedding $g_j^{\prime}(f(Y_{j,k}))$. The higher classification accuracies (Table 1,2) support (b). Moreover, we showed that we learn an aligned shared latent space out of the encoder (Figure 3). This also suggests that the shared latent space is separated from subject-dependent signals and captures the key structure of the common stimuli, further supporting (b). The experiment of untrained cross-subject translation and the additional classification experiment on handling new subjects we showed in responses (copied below) support (c).
> >
> >     Our MRMD-AE model employs a single encoder for all subjects. We can project the new subject’s data onto the shared latent space and proceed with the low-dimensional embedding of the new data. Then we have the option to pass the data through any of the existing decoders and perform the translation task, which projects the new subject’s data onto another subject’s manifold.
> > We have some preliminary results examining the embedding of new subjects. We hold out one subject to use as the test subject, and train the model using all the other subjects. Then we apply the trained encoder to the held-out subject to get their embedding in the shared latent space. We then perform the classification task on the held-out subject’s embedding. We repeat the experiment for each subject in the leave one out fashion. For comparison, we used group PCA and SRM in the same manner. The MRMD-AE achieves $0.681\pm0.05$, outperforming SRM(0.592\pm0.016) and PCA (0.595\pm 0.012).
> >
> >     In the classification tasks, the embeddings that we performed classification on are all from unseen time points that are used as testing sets. In Table 3, we showed that although not trained for cross-subject translation, we can project the unseen timepoints of one subject to another subject’s fMRI space. These support point (d). Then in Figure 4 we show that our framework provides flexibility to specialize in cross-subject translation, further supporting (d).
> >
> > 3. With the common encoder and the applied alignment penalty, we can learn a subject-independent manifold despite batch effects due to collection differences and individual differences in subjects, as shown in Figure 3. The cross-subject batch effects or individual differences are accounted for by learning a more aligned shared latent space: denoising the parts of the signal that are due to batch effects to find the commonalities in the signal.
> >
> >     Due to the presence of individual differences in subjects, we can apply the manifold learning algorithm PHATE to each subject’s fMRI data and learn individual manifolds. These individual PHATE manifold embeddings are used in individual decoders, at the $g_j^{\prime}$ layer to apply the manifold regularization penalty.
> >
> > 4. In Figure 2, we showed that the MRMD-AE generated manifold embedding closely match the ground-truth subject-specific PHATE embedding, confirming that $g_j^{\prime}$ introduces the subject dependent batch effect back, approximates the per-subject PHATE manifold and allows for the following reconstruction by the rest of the decoder layers. The  $g_j^{\prime}$ layer achieves the manifold embedding, then the $g_j^{\prime\prime}$ layers reconstructs the fMRI data from the manifold embedding. In Table 3, we reported the  low reconstruction errors contributed by $g_j^{\prime\prime}$.
> >
> > 5. We would like to thank the reviewer again for carefully examining our paper. We have corrected the typos.
> >
> > We have updated the manuscript in the modified submission. The current pdf reflects most if not all the discussions prior to the submission update deadline.

---

### Official Review · Reviewer_gwm6 · 2021-11-04

**Correctness:** 4
**Technical Novelty And Significance:** 3
**Empirical Novelty And Significance:** 3
**Recommendation:** 6
**Confidence:** 5

**Main Review:**

Strengths:
1.	Apt controls and baselines are employed throughout. Ablations convey the utility of each component loss function employed to train the MRDM-AE model.
2.	All components of MRD-AE modeling framework seem well motivated.
3.	Thorough comparisons are employed with multiple datasets
4.	The paper is very clearly written and all analyses seem well-executed.

Comments/Critiques:
1. Some results are not particularly exciting, for instance using the proposed MRDM-AE  framework does not result in a major improvement over a standard manifold regularized AE atleast in terms of stimulus decodability.
2. It would have been useful to compare cross-subject alignment capabilities of the MRDM-AE framework against existing popular techniques for aligning cross-subject fMRI data like hyperalignment.
3. Have the authors tried to assess extendibility/generalization to new subjects? For instance, once the MRDM-AE is trained, the authors could get the embeddings from a new subject and study stimulus decoding performance in this new scenario? In general, how could the proposed autoencoder framework be extended to a new subject? Do the authors envision this would involve simply training a new subject-specific decoder?
4. How much does adding the translation penalty improve cross-subject translation in terms of correlation?
5. It would be nice to also have some statistical significance analysis to assess whether the improvement of MRMD-AE over the dominant shared response modeling framework (SRM) is significant or not.
6. It might also be useful to critically assess whether subject-specific decoders indeed capture individual variations or are they largely just capturing a common signal. For instance, are the reconstructed signals from each decoder truly subject-specific, in the sense of correlating best with the same subject’s signal than any other?


**Summary Of The Paper:**

This paper proposes a neural network-based modeling strategy to learn a common latent space from multi-subject fMRI data. In addition to capturing a useful common latent space, the proposed technique is further able to disentangle common representational patterns from subject-specific variations through the use of subject-specific decoders. The authors impose meaningful and desirable priors on the latent embedding, like geometric regularization and cross-subject embedding alignment. Unlike other manifold learning techniques, the proposed deep neural network modeling framework lends itself well to extendibility to new data (stimuli) since the PHATE embeddings are only required at training time in the geometric regularization loss. The proposed framework is tested on two large fMRI datasets and an improved stimulus decoding (from the shared space) and cross-subject translation accuracy is achieved over competitive baselines.

**Summary Of The Review:**

This paper presents a valuable methodological contribution to multi-subject fMRI data analyses. The merits of the proposed technique are well supported by the experiments and results.

---

> ### Author Response · Authors · 2021-11-18
> **Response to Reviewer gwm6**
>
> Thank you for the encouraging comments and insights.
>
> On results comparing with individual MR-AE: First of all, manifold regularized AE is not standard in the field. To our knowledge, it has been proposed once in [Duque et al.]  and we are the first to propose it to fMRI data. It is one innovation of our paper to learn low-dimensional manifold from fMRI data that can extend to new timepoints. We extend the MR-AE to a multi-decoder architecture which grants unique properties of an aligned shared latent space that capture the common stimuli signal shared by subjects and can be used on fMRI from a group of subjects. Across datasets and tasks, MRMD-AE outperforms comparison methods.
>
> On comparing against hyperalignment: We compared MRMD-AE with the shared response model (SRM) as an existing popular technique for aligning cross-subject data instead of hyperalignment because SRM learns a common space in low-dimensions whereas hyperalignment learns a common space at the voxel level, without dimensionality reduction. Given that MRMD-AE learns a dimensionality-reduced common space, we feel that SRM is the most appropriate comparison.
>
> On handling new subjects: First, our MRMD-AE model employs a single encoder for all subjects. We can project the new subject’s data onto the shared latent space and proceed with the low-dimensional embedding of the new data. Then we have the option to pass the data through any of the existing decoders and perform the translation task, which projects the new subject’s data onto another subject’s manifold.
> We have some preliminary results examining the embedding of new subjects. We hold out one subject to use as the test subject, and train the model using all the other subjects. Then we apply the trained encoder to the held-out subject to get their embedding in the shared latent space. We then perform the classification task on the held-out subject’s embedding. We repeat the experiment for each subject in the leave-one-out fashion. For comparison, we used group PCA and SRM in the same manner. On the Sherlock early visual ROI data and indooroutdoor label, our MRMD-AE achieves classification accuracy of $0.681\pm0.05$, outperforming SRM($0.592\pm0.016$) and PCA ($0.595\pm 0.012$).
>
> On translation penalty impact on correlation: Since the translation loss penalty is constructed by the MSE as in Eq.(2) more significant improvement was observed in the comparison of translation MSE. The correlation is still improved (0.2280 with translation penalty vs 0.2275 of PCA), but less significant. We can design penalties to encourage the correlation as well, it will be a task-specific modification.
>
> On statistical significance: Statistical significance is normally not reported in machine learning for improvements of one neural network over another. Most formulations of this kind of result would likely yield extremely low, yet meaningless p-values. There is no studied null model for randomized neural network performance to say that a performance difference is significant or not. However, we have shown that we improve stimuli classification on different ROIs of both Sherlock and Forrest datasets.
>
> Here is the anonymous link to our code: https://drive.google.com/drive/folders/1oMrZgZzo-wbZ6Kp_SsIlGG_7-_pxA8tK?usp=sharing

---

### Official Review · Reviewer_cq7U · 2021-11-08

**Correctness:** 4
**Technical Novelty And Significance:** 1
**Empirical Novelty And Significance:** 1
**Recommendation:** 3
**Confidence:** 5

**Main Review:**

## Strengths

-   An interesting idea of constraining the embedding space with an existing non-generalizable manifold learning solution.
-   A mostly-clear exposition and a well-written paper.


## Weaknesses

-   The paper mostly reuses existing ideas and thus has a limited technical novelty. Multi-loss optimization.
-   The experiments are performed on a single ROI time courses, which diminishes the value of the work. The multivariate nature of fMRI is not taken into account and the experiments have mostly toy-example nature.
-   Focus on sensory ROIs excludes chances of capturing semantic information encoded by the brain in he fMRI signal and thus devalues presented work for the neuroimaging community.


## Questions to authors

1.  Since the decoders are subject specific and training is happening on a few subjects simultaneously, it is unclear how a new subject is handled. Which $g'_i$ decoder is used in this case?
2.  The input sample in the experiments is not specified making impossible to evaluate how close to fMRI signal is the input. Is this a window of the univariate ROI time-course per stimulus marker?

Information about what is the input data is only available in the supplement and thus difficult to obtain, which caused confusion in my case.


**Summary Of The Paper:**

The papers presents a multi-decoder autoencoder model with an objective function encouraging representations of stimuli-related fMRI activity similarly across subjects and events. The main goal is to obtain an encoder that learns a consistent and reusable mapping from fMRI space to representation space.


**Summary Of The Review:**

Two main concerns: limited technical novelty and limited value for neuroimaging. My recommendation is based on these limitations.

---

> ### Author Response · Authors · 2021-11-18
> **Response to Reviewer cq7U**
>
> Thank you for providing the feedback and offering us the opportunity to clarify some important aspects.
>
> On the technical novelty of our work: To the best of our knowledge we are the first to propose a single-encoder-multiple-decoder network to learn latent space of high-dimensional fMRI data jointly from a group of subjects. The multi-decoder architecture together with a new alignment loss penalty allows for separating common information from subject-dependent variation and noise. In addition, we propose a geometric regularization such that the embedding maintains the data geometry. Finally, our model can generalize to new time points and new subjects, which are all novel contributions. Crucially, this multiple-decoding scheme’s main purpose is to allow a single encoding into a common embedding space to be learned in an end-to-end fashion. At deployment on a new subject’s data, a decoder is not necessarily needed. We can project it to the shared latent space. The novelty of this approach leverages this basic idea: namely that building this common embedding space on a large enough sample of subjects will enable generalization to new ones.
>
> On the fMRI data and ROI selection used in the experiment,  it appears there may be some misunderstanding. We kept the timeseries for each voxel in an ROI, such that the data is a multivariate measure of regional activity. For example, the early-auditory ROI contains 1018 voxels, so the data is 1018-dimensional. The Sherlock dataset has 1976  time points, each time point is a 1018-dimensional vector.  Our experiments were conducted on both sensory and higher-order ROI. The PMC (Posterior Medial Cortex) ROI is a higher-order ROI that goes beyond the sensory information to more semantic information, particularly memory (See ref. Chen et al., 2017) . Our proposed model performs well in all ROIs.
>
> On handling a new subject: First, as described above, our MRMD-AE model employs a single encoder for all subjects, thus is subject-agnostic and is a central novelty of our approach. We can project the new subject’s data onto the shared latent space and proceed with the low-dimensional embedding of the new data. Then we pass the data through any of the existing decoders and perform the translation task, which projects the new subject’s data onto another subject’s manifold, if needed.
>
> Here is the anonymous link to our code: https://drive.google.com/drive/folders/1oMrZgZzo-wbZ6Kp_SsIlGG_7-_pxA8tK?usp=sharing

---

### Decision · Program_Chairs · 2022-01-20

**Decision:**

Reject

**Comment:**

In this paper, the author present a method for learning a shared latent space between the fMRI activity of multiple individuals processing the same stimulus. The method consists of an auto-encoder with a single encoder and subject-specific decoders which is specifically regularized to decouple common and shared representations. This paper generated a lot of discussion between the reviewers and the authors, as well as between the reviewers. In light of these discussion, I cannot recommend acceptance at this point, as the paper is not ready. The main concerns were (1) about how the results and improvement are evaluated statistically, (2) that the baselines chosen were not strong enough and did not include existing approaches (neural or non-neural) and relatedly (3) that the paper was not framed correctly within the existing literature on finding shared spaces between participants, which would help with determining and understanding the novelty of the proposed approach. Some other smaller points were made by the reviewer can also strengthen the paper for a future submission in a neuroscience or machine learning venue.